# A novel viral RNA detection method based on the combined use of trans-acting ribozymes and HCR-FRET analyses

**Leonardo Ferreira da Silva**[1○]**, Aisel Valle Garay**[2○]*****, Pedro Felipe Queiroz**[1]**, Sophia Garcia de Resende**[1]**, Mayna Gomide**[1]**, Izadora Cristina Moreira de Oliveira**[2]**, Amanda Souza Bernasol**[2]**, Anibal Arce**[3]**, Liem Canet Santos**[2]**, Fernando Torres**[4]**, Ildinete Silva-Pereira**[5]**, Sonia Maria de Freitas**[2]**, Cíntia Marques Coelho**[1]*****

1 Laboratory of Synthetic Biology, Department of Genetics and Morphology, Institute of Biological Science, University of Brasília (UnB), Brasília, Federal District, Brazil, 2 Laboratory of Molecular Biophysics, Department of Cell Biology, Institute of Biological Sciences, University of Brasília (UnB), Brasília, Federal District, Brazil, 3 Institute for biological and medical engineering, Pontificia Universidad Católica de Chile, Santiago de Chile, Chile, 4 Laboratory of Molecular Biology, Department of Cell Biology, Institute of Biological Sciences, University of Brasília (UnB), Brasília, Federal District, Brazil, 5 Laboratory of Molecular Biology of Pathogenic Fungi, Department of Cell Biology, Institute of Biological Sciences, University of Brasília (UnB), Brasília, Federal District, Brazil

○ These authors contributed equally to this work.
* cintiacoelhom@unb.br (CMC); aiselvalle@gmail.com (AVG)

**Data Availability Statement:** All data and results of the figures can be accessed via the Origin program, available in the Figshare public repository: https://doi.org/10.6084/m9.figshare.26877172.

## Abstract

The diagnoses of retroviruses are essential for controlling the rapid spread of pandemics. However, the real-time Reverse Transcriptase quantitative Polymerase Chain Reaction (RT-qPCR), which has been the gold standard for identifying viruses such as SARS-CoV-2 in the early stages of infection, is associated with high costs and logistical challenges. To innovate in viral RNA detection a novel molecular approach for detecting SARS-CoV-2 viral RNA, as a proof of concept, was developed. This method combines specific viral gene analysis, trans-acting ribozymes, and Fluorescence Resonance Energy Transfer (FRET)-based hybridization of fluorescent DNA hairpins. In this molecular mechanism, SARS-CoV-2 RNA is specifically recognized and cleaved by ribozymes, releasing an initiator fragment that triggers a hybridization chain reaction (HCR) with DNA hairpins containing fluorophores, leading to a FRET process. A consensus SARS-CoV-2 RNA target sequence was identified, and specific ribozymes were designed and transcribed *in vitro* to cleave the viral RNA into fragments. DNA hairpins labeled with Cy3/Cy5 fluorophores were then designed and synthesized for HCR-FRET assays targeting the RNA fragment sequences resulting from ribozyme cleavage. The results demonstrated that two of the three designed ribozymes effectively cleaved the target RNA within 10 minutes. Additionally, DNA hairpins labeled with Cy3/Cy5 pairs efficiently detected target RNA specifically and triggered detectable HCR-FRET reactions. This method is versatile and can be adapted for use with other viruses. Furthermore, the design and construction of a DIY photo-fluorometer prototype enabled us to explore the development of a simple and cost-effective point-of-care detection method based on digital image analysis.

**Funding:** This study received financial support from the following funders: - Financial support to C.M.C. from Just One Giant Lab (JOGL), France. - Financial support to F.T. from Conselho Nacional de Desenvolvimento Científico e Tecnológico (CNPq, process 406375/2022-4), Brazil. - Financial support to S.M.F. from Conselho Nacional de Desenvolvimento Científico e Tecnológico (CNPq, process 305819/2022-4), Brazil. - Financial support to S.M.F. from Fundação de Apoio à Pesquisa do Distrito Federal (FAPDF, Project No 00193-00000781/2021-19), Brasília, DF, Brazil, that include scholarships to L.C.S., A.V.G., I.C.M.O., and A.S.B. - Decanato de Pesquisa e Inovação e Decanato de Pós-Graduação (DPI/DPG) University of Brasília, Molecular Biology program to P.F.Q., and with Coordenação de Aperfeiçoamento de Pessoal de Nível Superior (CAPES) to L.F.S. - Financial support to C.M.C. from Comitê de Pesquisa, Inovação e Extensão de combate à COVID-19, Decanato de Pesquisa e Inovação Decanato de Extensão from University of Brasília (Project 7174/FUB/EMENDA/DPI/COPEI-UnB). - Financial support to S.M.F. from Comitê de Pesquisa, Inovação e Extensão de combate à COVID-19, Decanato de Pesquisa e Inovação Decanato de Extensão from University of Brasília (Project 7166/FUB/EMENDA/DPI/COPEI-UnB). The funders had no role in study design, data collection and analysis, decision to publish, or preparation of the manuscript.

**Competing interests:** The authors have declared that no competing interests exist.

## Introduction

Various methodologies are available for the detection and identification of viral RNA, with a predominant reliance on enzymatic processes involving RNA\DNA polymerases [1–4]. However, the use of enzymes presents logistical challenges including transportation to collection sites, and the need for specialized personnel and equipment for the analytical procedure. Particularly in countries and regions with vast geographical expanses, the implementation and distribution of these methodologies encounter obstacles, especially in regions with limited healthcare infrastructure.

Alternative diagnostic approaches, such as lateral flow immunochromatography, have shown effectiveness in widespread testing due to their affordability, simplicity, and rapid results. However, according to the Centers for Disease Control and Prevention (CDC), antigen tests are less sensitive compared to counterparts acid nucleic based and on reverse transcription quantitative polymerase chain reaction (RT-qPCR) [5]. Recent investigations highlight the inherent uncertainty and unreliability of antigen tests when confronted with a low viral load [6]. Contrarily, RT-qPCR recognized as the gold standard for pathogen detection, involve the use of reverse transcriptase enzyme and a real-time quantification probe with remarkable sensitivity and specificity for virus in the early stages of infection. It is the primary test employed for emerging viruses, such as SARS-CoV-2.

Application of ribozymes has primarily been proposed in therapeutic contexts [7], offering the advantage of generating fragments conducive to probe functionality [8, 9]. Therefore, protein enzyme-free strategies for detecting specific nucleic acid targets using cleaving ribozymes could be explored. Hammerhead ribozymes are small RNAs (~50–150 nt) capable of catalyzing their own self-cleaving and sequence-specific endonucleolytic cleavage. In solution, their secondary structure adopts a 'Y'-shaped fold, with three double helices (helix I to III) intersecting at a three-way junction that contains the catalytic core of 15 highly conserved nucleotides [10]. Hammerhead ribozymes exhibit stability during transport, storage, and manipulation. In the early 2000s, ribozymes were employed for detecting the hepatitis C virus. This detection method involved the binding of ribozymes to the viral genome using substrate one, which was cross-linked with biotin at its 3' end, and substrate two, cross-linked with 5'-fluorescein. The detection process used a streptavidin-coated microtiter plate and involved enzyme-dependent signal amplification using an anti-fluorescein antibody conjugated with alkaline phosphatase [11].

Additionally, oligonucleotide probes have been designed to detect specific sequences of interest by fluorophores [12–14]. A conceptual innovation is the signal amplification achieved through the hybridization chain reaction (HCR) technique. The concept of HCR was introduced by Dirks and Pierce in 2004 [15]. In HCR, stable DNA monomers are assembled only in the presence of a target DNA or RNA fragment. This process involves two stable DNA hairpins, containing complementary regions, which coexist in solution until the initiator strands trigger a hybridization cascade, forming nicked double helices. The molecular weight of HCR products varies inversely with initiator concentration, allowing DNA to act as an amplifying transducer for biosensing applications [15, 16]. Despite its advantages, HCR faces challenges that need to be addressed. Optimizing reaction conditions for consistent performance, improving the speed of the HCR process, and developing robust protocols for multiplexed detection are key areas needing advancement.

To optimize it and broaden the methodology's applicability, a Fluorescence Resonance Energy Transfer (FRET) reaction has been incorporated into the HCR approach. The FRET operates based on the transference of energy fluorescence between donor-acceptor molecular pairs, making it a highly efficient method for diagnostic applications [16–18]. This

methodology HCR-FRET has been applied to identify and image mRNA in situ using fluorescence microscopy, and to detect tumor-related mRNA in single cells and tissue sections with high sensitivity [19, 20]. This ability is promising for early cancer diagnosis by distinguishing between cancer and normal cells [16]. Thus it, offers the advantage of single-step execution and specific identification of the target molecules using a fluorophore-labeled molecular probe and smartphones [20].

During the emergence of the SARS-CoV-2 pandemic, millions of fatalities and extensive disruptions in health systems and societal domains occurred [21–23]. It was crucial during the pandemic to implement mass application diagnostic tests capable of detecting the virus in its early stages, along with rapid and decentralized non-pharmacological control measures [24–26]. Our approach aims to introduce a novel modality for viral RNA detection, using SARS-CoV-2 as a proof of concept. This involves the use of trans-acting ribozymes for the recognition and cleavage of viral RNA, complemented by DNA hairpins harboring containing Cyanine 3/Cyanine 5 (Cy3/Cy5) fluorophores to initiate a HCR based on HCR-FRET for the amplification of signals.

Additionally, to exploring the feasibility of utilizing a low-cost device for FRET detection using the proposed diagnostic molecular method, an image-based *Do-It-Yourself* (DIY) spectrofluorometer was prototyped. In this device the fluorescence quantification was performed using a digital image-based method (DIB), which is widely adopted for its ease of use, portability, speed, precision, accuracy, and cost-effectiveness. Additionally, DIB offers potential for *in situ* analysis by point-of-care testing device [27–29]. In the DIB method an image capture with cameras or smartphones is converted into measurable data that is correlated with the concentration of the analyte [27, 28]. Finally, the goals of this propose is to develop an easy-to-use detection methodology that does not rely on thermo-sensitive enzymatic requirements nor on specialized instrumentation or highly skilled personnel.

## Material and methods

### Identification of genomic targets

A SARS-CoV-2 genomic sequence for ribozymes cleavage was initially selected considering a previously successful used target [30]. This region was analyzed in an alignment of approximately 16,000 SARS-CoV-2 genome sequences available in the NCBI database (https://www.ncbi.nlm.nih.gov/genbank/sars-cov-2-seqs/) using the MAFFT open access software (https://mafft.cbrc.jp/alignment/server/). The default parameters were employed, except for the 'output format', which was chosen as clustal format sorted, and the 'strategy', which was set to FFT-NS-2. Data visualization was performed using Jalview software (https://www.jalview.org/). This viral region was also aligned with sequences from other RNA viruses, close relatives of SARS-CoV-2, such as those from SARS and MERS families, to exclude sequences that are also consensual for these other viruses. Then, a sequence of 400 pb, in ORF1ab, was identified as consensus and potentially specific for SARS-CoV-2 since it was different from the sequences found in other RNA viruses and, therefore, it was chosen as the target sequence of the ribozymes.

### Rational design of ribozymes and DNA reporter harpin molecules

Extended hammerhead-type ribozymes were designed using the RiboSoft program according to the default parameters, except for the parameter "target environment", which was selected: *in vivo*, Homo sapiens (taxonomy 9606, species 9606) using assembly GRCh38.p12 and the specificity method that was selected "cleavage"27 (https://ribosoft2.fungalgenomics.ca/). The RNA sequence of 400 nt was used as the target template and three hammerhead-type ribozymes, called 1, 2 and 3, were selected based on cleavage site, specificity, accessibility, and

**Table 1. DNA and RNA molecules sequences used in this study.**

| Name | Sequence (5' → 3') | nt |
|---|---|---|
| ORF1ab | UUAGAUAUAUGAAUUCACAGGGACUACUCCCACCCAAGAAUAGCAUAGAUGCCUUCAAACUC AACAUUAAAUUGUUGGGUGUUGGUGGCAAACCUUGUAUCAAAGUAGCCACU**GUACAGUC \| UA AAAUGUCAGAUGU**AAAGUGCACAUC***AGUAGUC \| UUACUCUCAGUUUUG***CAACAACUCAGAGU AGAAUCAUCAUCUAAAUUG***UGGGCUCAAUGUGUC \| CAGUUACACAAUGAC***AUUCUCUUAGCU AAAGAUACUACUGAAGCCUUUGAAAAAAUGGUUUCACUACUUUCUGUUUUGCUUUCCAUGCA GGGUGCUGUAGACAUAAACAAGCUUUGUGAAGAAAUGCUGGACAACAGGGCAACCUUACAAGC UAUAGCCUCAGAGUUUAGUUCCCUUCCAUC | 400 |
| *Ribozyme 1* | CAAAACUGAGAAUAGUAACUGAUGAGUC GCUGAAAUGCGACGAAACUACU | 50 |
| *Ribozyme 2* | GUCAUUGUGUAAUAACUGCUGAUGAGUCGCUGAAAUGCGACGAAACACAUUGAGCCCA | 58 |
| **Ribozyme 3** | ACAUCUGACAAAUUUUUACUGAUGAGUCGCUGAAAUGCGACGAAACUGUAC | 51 |
| Target sequence 1 | AGTTTAGTTCCCTTCCAT | 18 |
| H1 (first set) | ATGGAAGGGAACTAAACTTGTGATAGTTTAGTTCCC | 36 |
| H2 (first set) | AGTTTAGTTCCCTTCCATGGGAACTAAACTATCACA | 36 |
| Target Sequence 2 | GATGTAAAGTGCACATCA | 18 |
| H1 (second set) | TGATGTGCACTTTACATCTGTGATGATGTAAAGTGC | 36 |
| H2 (second set) | GATGTAAAGTGCACATCAGCACTTTACATCATCACA | 36 |

The ribozymes complementary sequences within the 400 nt ORF1ab region are highlighted as follows: **Bold** indicates the sequence for ribozyme 3, ***Bold-Italic*** represents the sequence for ribozyme 1, and <u>***Bold-Italic-Underline***</u> denotes the sequence for ribozyme 2. Vertical lines in the sequences indicate their respective cleavage sites.

structure scores. To select a region of the 400 nt RNA target to be complementary to the first hairpin reporter DNA molecule (H1) an analysis of the pair probability of every nucleotide in windows of 18 nt was performed, finding the sequence with the highest probability of being single stranded in the ensemble. This sequence was selected, and the first set of hairpin DNA reporters (H1 and H2) was designed to be complementary to it. A second H1 and H2 pair was designed to be complementary to the RNA fragment obtained after the action of ribozymes 1 and 3. Both sets were designed according to Ang and Yung [17]. These molecules were then evaluated using NUPACK software [31] (http://www.nupack.org/) according to the following parameters: DNA, Compute Melt: ON, Temp min: 25 ˚C, Increment: 6 ˚C, Temp Max: 37 ˚C, Na 0.75 M, 3 strands and max complex size 3, [molecules]: 100 nM for each one. They were synthesized by IDT (Integrated DNA Technologies Coralville, Iowa, USA), H1 containing Cy3 fluorophore at 5' end and H2 with the Cy5 fluorophore positioned at the 3' end. The 400 nt RNA target sequence, all the ribozymes and DNA hairpins sequences are presented in Table 1. The RiboSoft obtained scores are shown in Table 2.

## Evaluation of ribozyme activity

The selected 400 nt viral RNA target and the three hammerhead-type ribozyme sequences were synthesized as dsDNA sequences attached to a T7 promoter by GenScript (GenScript, Piscataway, New Jersey, USA). These DNA molecules were inserted in the cloning vector pBluescript SK (+) by the inFusion HD EcoDry Cloning Kit (Takara Bio, San Jose, California, USA), and then transformed into thermal-competent bacteria (XL-10 Gold). Four different vectors were obtained, pBluescript SK (+)_Rz1, pBluescript SK (+)_Rz2, pBluescript SK (+) _Rz3, and pBluescript SK (+)_target. Then, the RNA target and the ribozymes were transcribed separately using the MEGAshortscript T7 Transcription Kit (Invitrogen, Waltham, Massachusetts, USA), according to the manufacturer's instructions. Transcripts were quantified, analyzed by agarose gel electrophoresis, and incubated at -80 ˚C until use. To assess ribozyme cleavage activity, reactions were performed as described by Tang et al., 1994 [32]. Briefly,

**Table 2. Parameters obtained by RiboSoft software for each one of the ribozymes used in this study.**

| Ribozyme | Specificity Score | Accessibility Score | Structure Score |
|---|---|---|---|
| 1 | 5081.81 | 7.1 | 13.38 |
| 2 | 1534.499 | 9.6 | 33.009 |
| 3 | 2352.182 | 5.9 | 15.86 |

the 400 nt target RNA molecule and each one of the three ribozymes were separately mixed in a 1:10 ratio in the ribozyme reaction buffer containing 20 mM MgCl2 (Dinâmica, Indaiatuba, São Paulo, Brazil) and 50 mM Tris-HCL, pH 8.0 (Promega, Madison, Wisconsin, USA). The solution was incubated for 2.5 h at 50 ˚C. All assays were performed separately in technical duplicates and independent experiments, considering negative control, the absence of ribozymes. Ribozymes, the 400 nt target RNA, and cleaved products were analyzed by 2% agarose gel electrophoresis. Aiming to enhance the efficacy of the final testing strategy the ribozymes were subjected to dried, prior to assessing their activity, using a Savant SPD1010 SpeedVac Concentrator System (Thermo Scientific, Waltham, Massachusetts, U.S.A.) for 20 minutes at 50 ˚C. Subsequently, the ribozyme activity assay was conducted as described. Finally, the decrease in the duration of ribozyme activity was evaluated by sampling at intervals during the incubation period, specifically at 10, 30, 60, 90, and 150 minutes. The resulting products were then analyzed through 2% agarose gel electrophoresis. Horizontal agarose gel electrophoresis was performed using RNase-free low melting agarose 2% (w/v) in a 15 x 15 cm UV-transparent tray system with PowerPac™ Basic Power Supply (Bio-Rad Laboratories, Inc., Hercules, California, USA). The electrophoresis was conducted with 1x Tris/Acetic Acid/EDTA (TAE) nucleic acid electrophoresis buffer solution at pH 8.3, at a constant 80 Volts. Subsequently, the gels were stained with 0.01 mg/mL of ethidium bromide [33].

## HCR-FRET-based signal detection and amplification test

Absorption and fluorescence spectra of H1-Cy3 and H2-Cy5 were measured using a UV/Visible Spectrophotometer Jasco V-530 (Jasco Analytical Instruments, Tokyo, Japan) and a SpectraMax M3 plate reader SpectraMax M3 (Molecular Devices, San Jose, California, USA), respectively. H1-Cy3 and H2-Cy5 were excited at their respective absorption maxima of 547 nm and 647 nm, respectively. The spectra were obtained with three replicates. To evaluate the HCR-FRET signal, the hairpin molecules (H1 and H2) were diluted separately in a DNA hybridization buffer containing 5X saline sodium citrate (SSC) (Sigma-Aldrich) with the addition of 0.1% Tween-20 (Sigma-Aldrich) for the initial assays and without Tween after the optimization assays. Then, the hairpin molecules were heated to 95˚C for 5 min and cooled to room temperature for 40 min to recover the original structure. The negative control corresponds to the sample without the target/initiator molecule. To set up the best parameters for the HCR-FRET assays, first they were performed with the 18 nt initiator DNA molecule synthesized by GenScript (Piscataway). Different concentrations of the 18 nt initiator DNA molecules diluted in 5X SSC with and without Tween were used at 6000 nM, 600 nM, 400 nM, and 200 nM final concentrations. These molecules were mixed with H1 and H2 (600 nM final concentration), and then the samples were transferred to a Greiner black 96-well, f-bottom reading microplate (Greiner Bio-One, Kremsmünster, Austria). Fluorescence spectra were measured at 27 ˚C in three different spectrofluorometers, K2 Multifrequency phase fluorometer (ISS, Champaign, Illinois, USA) attached to microwell plate reader K428, Varioskan LUX (Thermo Fisher Scientific, Waltham, Massachusetts, USA), and SpectraMax M3 multimode microplate reader. An 18 nt RNA initiator molecule synthesized by GenScript and diluted in

5X SSC buffer was also evaluated in HCR-FRET assays using the Varioskan LUX and Spectra-Max M3 multimode microplate reader.

To evaluate the detection method, the 400 nt RNA ribozyme cleavage products plus one set or two sets of DNA hairpins H1 and H2 molecules at a final concentration of 600 nM were mixed in a 2:1:1 ratio in a total volume of 100 μl of 5X SSC, as previously described [16]. As positive control the 18 nt RNA1 or/and RNA2 initiator molecules diluted in the ribozyme cleavage buffer was used. Fluorescence spectra were recorded at 27 ˚C immediately after the assay using the SpectraMax M3 multimode microplate reader. HCR-FRET signals of the fluorescence spectra were monitored with excitation at 547 nm and emission at 557–750 nm, corresponding to the excitation and emission wavelengths of donor and acceptor fluorophores, respectively. To evaluate the specific recognition on the test on viral RNA, two non-relevant DNA molecules (not related to the viral genome) were chemically synthetized (CS) and employed as substitutes for the target: CS-1 5′CCTCTCAAATAAGTGGCGCAGATGCTCT AG3′ and CS-2 5′CATT GGCGACGTCGCGCTTTCC 3′. The detection limit was established by varying concentrations of the RNA target, ranging from 0.5 nM to 600 nM, while maintaining a constant concentration of H1 and H2 variants. To evaluate the detection method-time limit, the 10 minutes ribozyme 1 cleavage products had the fluorescence spectra monitored after 0 minutes, 10 minutes, 40 minutes, and one hour with HCR-FRET signals of the fluorescence spectra analyzed as described above. All assays were performed separately, in duplicates on different experimental days.

## Design and manufacture of device

A 96-wells plate reader DIY Photo-fluorometer device was developed and manufactured using for fluorescence quantification method through digital images, known as DIB. These 3D-printable parts were designed using the computer-aided design (CAD) software, Solidworks® 3D CAD Premium 2016 x64 Edition (Dassault Systèmes, SolidWorks Corporation, Waltham, Massachusetts, U.S.A.). Each designed 3D model of components was exported in ".STL" file format for further slicing and setting of printing parameters into ".gcode" files compatible with 3D printers. This was accomplished using the free and open-source Orca Slicer v2.0.0 software (http://github.com/SoftFever/Orcaslicer). Additionally, this device was specifically designed and manufactured to enable fluorescence detection through imaging, using a professional digital camera mounted within a black dark chamber to capture photos of the 96-well plate.

The plastic device parts were manufactured using Fused Deposition Modeling (FDM) 3D printing technology with a K1 Creality 3D printer (Shenzhen Creality 3D Technology Co, Ltd., China, https://crealitystore.com.br), utilizing polylactic acid (PLA) filament in various colors. The 3D print slicer parameters included a 0.4 mm extrusion nozzle, a layer thickness of 0.2 mm, 3 walls each with a width of 0.42 mm, a 25% infill using a grill pattern, and variable printing speeds ranging from 60 to 500 mm/s, with tree-type supports. The printing process adhered to a high precision of ±0.1 mm, maintaining a printing bed temperature of 60 ˚C and a nozzle temperature of 220 ˚C. To minimize reflections and light interference within the dark chamber, black PLA filament was consistently used. After printing, surface smoothing and finishing were performed to remove imperfections and wrinkles using sandpaper and tweezers. Assembly of the components and parts was accomplished through cold fittings, employing plastic insert nuts, screws, and a special plastic adhesive to ensure a secure and effective assembly.

The device consists of several modules that offer flexibility in both configuration and operation. These modules include: a red, green, and blue (RGB) light-emitting diode (LED) module, an electronic control module, illuminated box for photography, user interface module with a 2x16 LCD display with i2c module and 3x4 matrix keyboard, passive Buzz, PCF8574 Digital I/

O Expander via I2C-bus by serial clock (SCL), serial data (SDA), and the control software. The LED module was designed with a 12V-DC SMD 5050 RGB LED strip as the light source, which can emit light in three primary colors red, green, and blue to emit quasi-monochromatic excitation light for the primary donor fluorophore. To ensure uniform light intensity on the samples plate, a compact modular illuminator unit was developed and manufactured using the RGB LED strip and diffuser plate to scatter light effectively, positioned at a 45° angle relative to the sample board. The LED module was electronically controlled via an Arduino nano V3 Atmega328 Ch340 microcontroller board, employing three TIP122 MOSFET transistors assisted by 220 Ω resistors for precise control over the red, green, and blue channels through Pulse Width Modulation (PWM). PWM output pulses ranged from 0 to 255, enabling the mixing of different PWM levels to generate a wide spectrum of colors. To generate a 5V voltage to power the Arduino Nano board and other components from a 12V source, an LM2596 voltage regulator into the control module was integrated. This regulator can supply up to 3A of direct current, with a nominal output range of 4.5 to 28V from an input range of 0.8 to 20V DC. The electronic circuit was designed using the open-source software Fritzing v0.9.9 (available at https://fritzing.org/) and mounting on a perforated prototyping board. All electronic components used during the prototyping phase of the device variants were sourced from specialized electronics stores. The control software was written on Arduino's Integrated Development Environment (IDE) (version 1.8.9) was used, as well as the Processing software, which are based on C programming language. This combination allowed us to efficiently program the Arduino Nano board to control the RGB LED strip.

Wavelength calibration for each RGB channel of LED was achieved using the spectrofluorometer Jasco FP-6500 (Jasco Analytical Instruments, Tokyo, Japan) considering only the fluorescence emission channel. To achieve this, the JASCO ETC-273T Peltier module was removed, and the RGB LED module was accommodated into compartment. The excitation optical channel in the spectrofluorometer was blocked to prevent the entry of excitation light signals into the compartment. Only the emission recording channel of the spectrofluorometer was used to capture the emission spectra of the RGB LED module through its different RGB channels. Additionally, an intensity attenuator physical filter was employed during readings to prevent detector saturation. Parameters used in the experiment included: an emission bandwidth of 5nm, a response time of 0.01 seconds, variable gain settings (low or high) paired with a physical intensity filter, spectral recordings spanning from 300 to 700 nm at a rate of 100 nm per minute, with a data interval of 0.5 nm.

## Detecting FRET with DIY Photo-fluorometer device

To assess the HCR-FRET signal using an image-based DIY Photo-fluorometer, five different concentrations of equimolar mixture of target 18 nt initiator RNA1 and RNA2 were tested. The RNA targets were incubated in 5X SSC without Tween, with 600 nM of H1 and H2, using the combination of two DNA hairpins H1/H2 sets. The concentrations tested were 50 nM, 100 nM, 200 nM, 300 nM, 600 nM, and 1200 nM. Additionally, a negative control was included where no target DNA initiator was added. Each assay was repeated four times for reliability. The FRET reactions were carried out as previously described and subsequently transferred to a Greiner black 96-well, F-bottom reading microplate. To confirm the FRET sinal, emission spectra were recorded from 557 to 750 nm with a data interval of 1 nm using a SpectraMax M3 multimode microplate reader with the following parameters: excitation wavelength 547 nm, 6 flashes per read from top, with photomultiplier tube (PMT) gain in automatic and without shaking. For optimal image capture, reactions were transferred to a Greiner white 96-well plate with a flat bottom to enhance photo contrast. During the image capture process, the user

selects the desired wavelength (Red, Green, or Blue) using the keyboard to activate the corresponding LED for sample excitation, based on the range established by LED calibration. The Arduino code associated with this process can be found in the supplementary information (S1 Appendix). Digital camera Canon EOS Rebel XS DS126191 equipped with a 24.1-megapixel CMOS (APS-C) sensor and DIGIC 8 image processor, paired with an EFS 18-55mm lens (Canon Inc., Taiwan), was used for image capture.

### Detecting and analyzing FRET from digital image

The images were captured as ".CR2" files and then converted to ".JPG" files with maximum quality using three progressive scans in Adobe Photoshop, version 12.0 software (Adobe Systems Incorporated, San José, California, U.S.A.), employing the ICC (International Color Consortium) algorithm Adobe RGB (1998). The image specifications were as follows: Depth 8 Bits/Channel, Size approximately 1750x2930 pixels (10.1 MP), Resolution 240 pixels/inch. For each capture using the DIY device, excitation was performed with green LEDs to emit FRET in the red spectrum region. The acquired images were analyzed using ImageJ v1.53e software, developed by the National Institutes of Health and written in Java (available at http://imagej. nih.gov/ij). The analysis were performed using the ReadPlate v3.0 plugin (https://imagej.nih. gov/ij/plugins/readplate *index.html*) or script within the ImageJ software environment, which simultaneously extracted from each well the Red (R), Green (G), and Blue (B) color values from the images. To achieve this, a grid of circles is used, with main circles centered on each well, surrounded by three ancillary small circles positioned outside. These ancillary circles are used in the blank correction algorithm to compensate for any variations in local light intensity. Corrected absorbance values (Acorr) are obtained by applying the blank suppression algorithm. Parameters were configured with a circle diameter set between 90 pixels on the wells, three ancillary circles considered for blank, a proximity factor of 1, and size factor of 0.577. The resulting data were saved in Excel format, exported to ".txt" files, and organized, analyzed and visualized using OriginPro® 2015 software (OriginLab Corporation, Northampton, Massachusetts, U.S.A.).

## Results

### Molecular mechanism for SARS-CoV-2 RNA detection method

The proposed method to detect SARS-CoV-2 viral RNA, as proof of concept, consists in several steps (Fig 1). In the first step the viral RNA genome is recognized and cleaved by hammerhead-type ribozymes (Fig 1A). The activity target sites for each of the chosen ribozymes must be on specific regions only to the SARS-CoV-2 genome sequences. For this purpose, the target sequence of SARS-CoV-2 was obtained from a previous study [30]. The target sequence is a 400 nt fragment within the ORF1ab region (YP_009724389.1, positions 11,459–11,858) corresponding to the non-structural protein 6 (Nsp6) (https://www.ncbi.nlm.nih.gov/gene/ 43740578). Nsp6 is a highly conserved membrane protein with six transmembrane domains, essential for viral replication [34, 35]. The alignment with 16,000 consensus sequences of SARS-CoV-2 achieved over 98% identity (https://doi.org/10.6084/m9.figshare.25922725), making the 400 nt fragment an ideal target for ribozyme design. Three ribozymes targeting this region were selected and their specificity were evaluated by aligning target sequences with the genomes of related coronaviruses (SARS-CoV, MERS-CoV, and other common human coronaviruses). The results revealed significant nucleotide sequence differences between SARS-CoV-2 and the other coronaviruses, indicating the specificity of selected ribozymes. The selected 400 nt ORF1ab viral target sequence is recognized and cleaved by the hammerhead-type ribozymes releasing an initiator fragment molecule, considered as an input for the

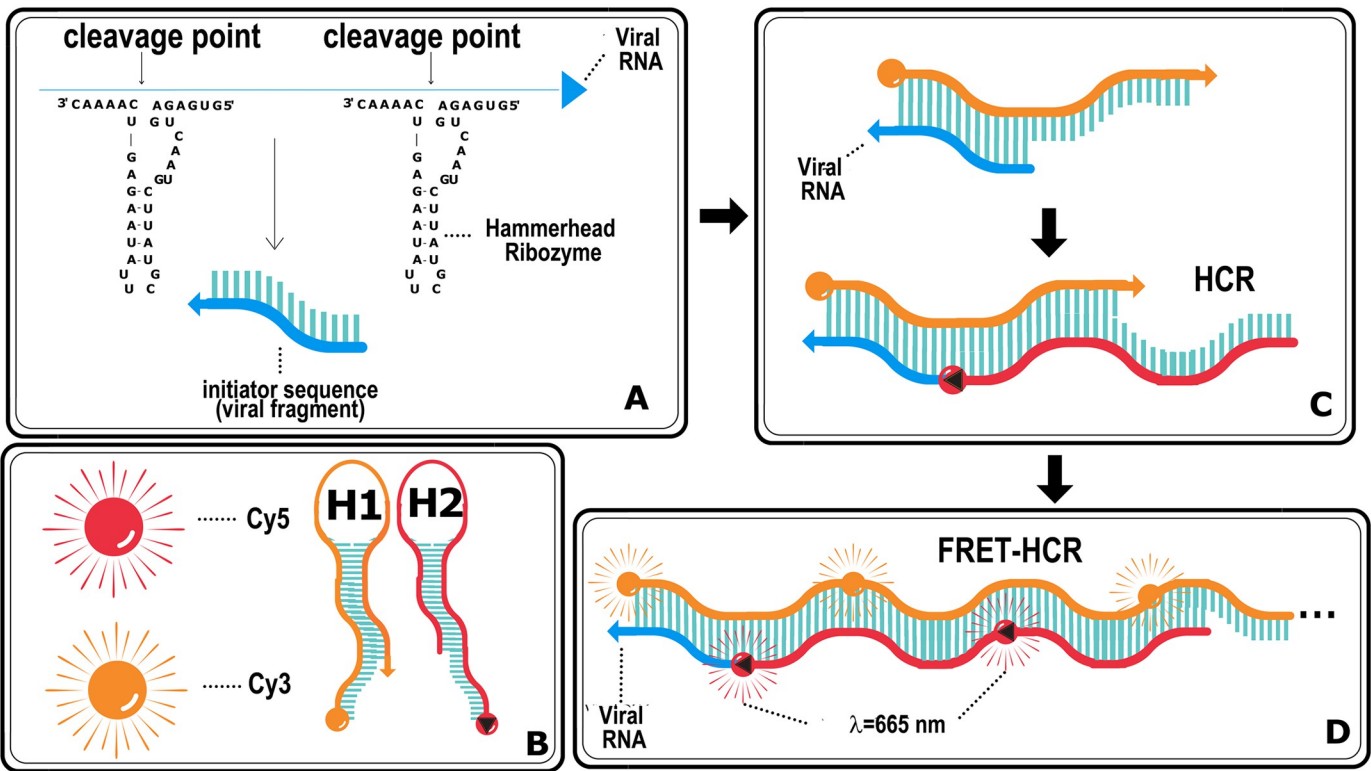

**Fig 1. Schematic representation of the main reaction steps of the diagnostic method.** Detection and cleavage of the target viral RNA by hammerhead ribozymes, resulting in the release of an initiator fragment (A). Schematic representation of two reporter DNA molecules containing different fluorophores: H1-Cy3 (Cyanine 3) and H2-Cy5 (Cyanine 5) (B). Binding of the initiator molecule to the H1 reporter DNA molecule, causing its opening and subsequent binding and opening of the H2 reporter DNA molecule through a hybridization chain reaction (HCR) (C). Formation of a polymer of concatenated H1 and H2 molecules, resulting in HCR-FRET that amplifies the detection signal (D).

detection method. Table 1 shows all ribozyme sequences and the 400 nt of the RNA target molecule used in this study.

The second part of the process is based on the complexation of this released initiator viral RNA fragment with two metastable DNA hairpin containing the donor (Cy3, Cyanine 3) and acceptor (Cy5, Cyanine 5) fluorophores (Fig 1B and 1C). According to absorption and emission fluorescence spectra of H1-Cy3 and H2-Cy5 (S1 Fig), the donor fluorophore could be excited at a maximum absorption wavelength of 547 nm, and the excitation energy is transferred to the acceptor resulting in an increase in the intensity of fluorescence emission at ~ 660–670 nm by FRET process. The first DNA hairpin (H1) opens upon interacting with the viral RNA initiator fragment through complementary binding to its toehold and stem sequence. Following this, the second DNA hairpin (H2) opens after hybridizing with the first H1, due to complementary binding with its extended loop and stem sequences. H1 and H2 were designed to position the two fluorophores close to each other, ensuring optimal conditions for the FRET reaction (Fig 1B). The third part is based on the increase in the fluorescence signal at the acceptor wavelength due to the concatenation of H1 and H2 DNA hairpin molecules forming a polymer. This polymerization chain reaction potentially allows the detection of small amounts of viral RNA (Fig 1C and 1D). For this proposed detection method, the fluorescence emission spectra are corrected by subtracting the spectra of the corresponding negative controls, which consist of HCR-FRET assays performed without the initiator fragment molecule. A positive control will always be included.

## HCR-FRET-based signal detection and amplification

The design of the first pair or set of the DNA hairpin reporter molecules (H1 and H2) was performed according to previous studies [17]. H1 was complementary to an 18 nt sequence of the 400 nt SARS-CoV-2 target RNA considered the region most likely to be in single-stranded structure. Then, the designed H1 and H2 pairs were evaluated by the NUPACK program. Among the designed molecules, for the selection of the H1-H2 pair, the best results were comparatively considered, including lower free energies in the structure of the probes and in the formation of the concatenated complex; higher concentrations in the formation of the Target-H1-H2 complex; and lower percentages of free probes. Based on the results, the first pair of hairpin reporter DNA molecules H1 and H2, containing Cy3 and Cy5 fluorophores positioned at the 5' and 3' end, respectively, with potential for binding with the initiator fragment was selected. HCR-FRET was detected in different independent experiments. Synthetic initiator DNA molecules of the same size as the H1 toehold (18 nt) were initially used to optimize the concentration of the target molecule and the buffer conditions.

For these initial assays, the DNA molecules were diluted in a 5X SSC buffer with 0.1% Tween-20 (SSCT), which was the HCR-FRET buffer used by Ang and Yung, 2016 [17]. It was hypothesized that this 18 nt DNA molecule would bind specifically to H1, promoting its opening and enabling subsequent binding of H2 (Fig 1C). For the HCR-FRET assay, three different concentrations of the 18 nt initiator DNA molecule were evaluated, 100 nM, 200 nM, and 600 nM (Fig 2A). All analyzed spectra were obtained using a K2 fluorometer and after subtracting the negative control spectrum (without DNA molecules). The spectra exhibited a decrease in signal at ~625 nm and an increase in intensity at ~670 nm as the concentration of 18 nt initiator DNA molecules increased (Fig 2A). These observations were consistent across three to five independent experiments using H1 and H2 at a concentration of 600 nM in 5X SSCT buffer. Further optimization assays included the removal of the Tween detergent on the reaction buffer. When comparing the results at 600 nM of 18 nt initiator DNA molecules in 5X SSCT and 5X SSC (in the absence of Tween) buffers, no significant differences were observed (Fig 2B). Therefore, subsequent experiments were performed using 600 nM of 18 nt initiator DNA molecules in SSC buffer (without Tween detergent).

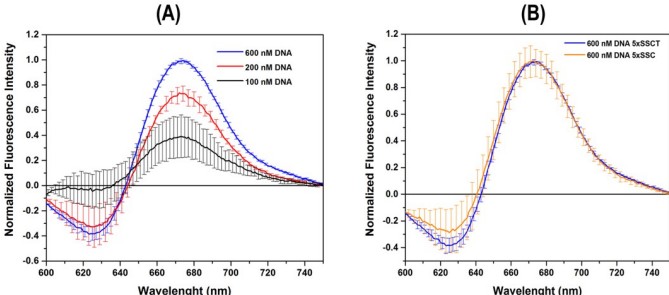

**Fig 2. Fluorescence spectra emission analysis of FRET between H1 and H2 on different DNA target concentrations.** Average fluorescence spectra emissions (n = 3 or five independent experiments) were measured using a K2 fluorometer. Spectra were obtained by subtracting the negative control spectrum (without initiator DNA molecules), resulting in a negative fluorescence signal at 625 nm. Experiments were conducted in 5X SSCT buffer with varying concentrations of 18 bp initiator DNA molecules. Three concentrations of initiator DNA were evaluated: 100 nM (black line), 200 nM (red line), and 600 nM (blue line) (A). Additionally, the effect of 5X SSCT (blue line) versus 5X SSC (magenta line) buffers was compared at an initiator DNA concentration of 600 nM (B). Standard deviations are represented by error bars. In all experiments, 600 nM of H1 and H2 hairpin fluorophore reporter DNA molecules from the first set, were used.

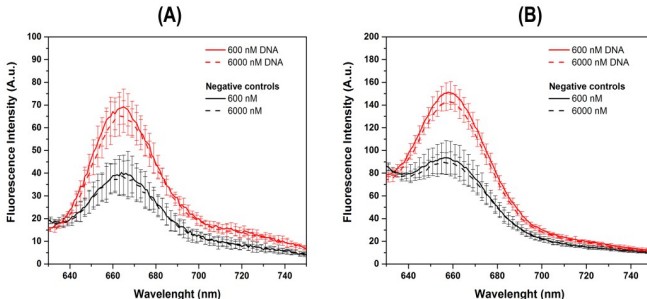

**Fig 3. Fluorescence spectra emission analysis of FRET between H1 and H2 on DNA target: Comparison between Varioskan LUX and SpectraMax M3 microplate readers.** Fluorescence emission spectra (n = three independent experiments) were recorded using the Varioskan LUX microplate reader (A) and the SpectraMax M3 multimode microplate reader (B) to assess HCR-FRET between H1 (600 nM) and H2 (600 nM) from first set. The experiments were carried out in 5X SSC buffer with different concentrations of 18 nt initiator DNA molecules (indicated by red lines) and negative controls without DNA target (black lines). Two concentrations of initiator DNA molecules were evaluated: 600 nM (solid lines) and 6000 nM (dashed lines). Standard deviations are represented with error bars. Bar graph plots displaying the maximal fluorescence intensity signals relative to the negative control signals are shown (C) with corresponding standard deviations. No significant differences were observed between concentrations using the same equipment. Additionally, the Varioskan exhibited a slightly increased sensitivity. Measurements were carried out considering the optimization of sample preparation, instrument calibration, standardization of plate reader parameters, and minimization of background fluorescence.

Additionally, to assess the HCR-FRET signal, experiments were conducted using two plate readers: the Varioskan LUX and the SpectraMax M3. Two different concentrations of DNA initiator molecules were evaluated: 600 nM and 6000 nM, with 600 nM of H1 and H2 from first set, used in each case. The spectra of negative control were acquired without initiator molecules. The results indicate that the HCR-FRET on the Varioskan LUX (Fig 3A) and Spectra-Max M3 (Fig 3B) was similarl to that obtained on the most sensitive spectrofluorometer, K2 (Fig 2), with maximum fluorescence emission at approximately 660 nm (Fig 3). The bar graph plots show that the ratio of fluorescence intensity signal to the negative control was 1.6 and 1.7 for the HCR-FRET assays at both concentrations, 600 nM and 6000 nM, performed on the Spectramax and Varioskan, respectively (Fig 3C). Therefore, the results indicate that HCR-FRET can be effectively detected at the lowest concentration (600 nM), despite differences in equipment sensitivity.

Due to the genome of the SARS-CoV-2 being RNA and not DNA, the next step was to perform the HCR-FRET assay using the 18 nt RNA fragment of the SARS-CoV-2 genome at concentration of 600 nM as initiator molecule, and 600 nM of H1 and H2 from first set, in 5X SSC buffer (Fig 4A). The results shown the occurrence of HCR-FRET like those obtained with the 18 nt DNA molecule, with fluorescence emission of the acceptor at ~ 660 nm, being ~1.6 higher than the negative control as observed in the bar graph plots (Fig 4B). The SpectraMax M3 was chosen for the further analysis. To confirm specific recognition of the DNA hairpins to related SARS-CoV-2 viral RNA sequences, assays were conducted using different concentrations (0.5–600 nM) of CS-1 22 bp (Fig 4C) and CS-2 30 bp (Fig 4D) DNA sequences unrelated to the viral genome. The CS-1 and CS-2 sequences of DNA fragments are shown in material and methods. Assays were performed with 5x SSC buffer and 600 nM of H1 and H2 DNA hairpins from first set. In this assay, a positive control of the first 18 nt SARS-CoV-2 viral RNA target was used. The results demonstrate the absence of FRET fluorescent emission in any concentrations of CS-1 or CS-2 DNA sequences unrelated to the viral genome, confirming the specificity of the designed H1 and H2 molecules of first to viral genome recognition.

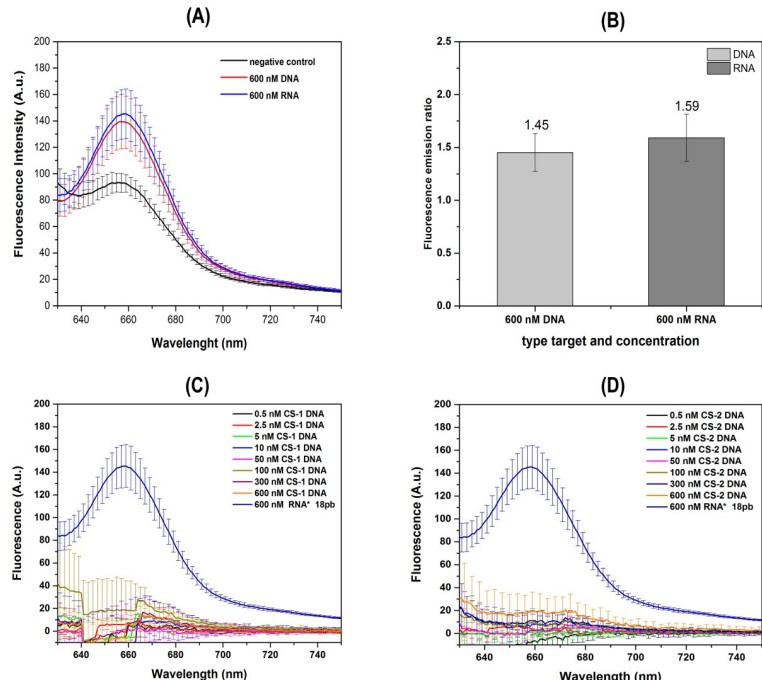

**Fig 4. Specific recognition of the DNA hairpins with related SARS-CoV-2 viral target DNA and RNA sequences.**
Average fluorescence intensity emission due to HCR-FRET between H1 and H2 with 18 nt initiator DNA (red line) and 18 nt RNA (blue line) in 5X SSC buffer with 600 nM of H1 and H2 from first set (A). Negative control without an initiator molecule is indicated as the black line. Bar graph plots show the maximal fluorescence intensity signals of the negative control signals with the 18 nt initiator DNA and RNA molecules with corresponding standard deviations (B). Specific recognition assay of the DNA hairpins by SARS-CoV-2 viral RNA sequences were conducted using different concentrations (0.5–600 nM) of CS-1 22 bp (C) and CS-2 30 bp (D) DNA sequences unrelated to the viral genome. The CS-1 and CS-2 sequences of DNA fragments are shown in material and methods. Assays were performed with 5x SSC buffer and 600 nM of H1 and H2 DNA hairpins. A positive control of first 18 nt SARS-CoV-2 viral RNA target was used (600 nM RNA* 18pb). Assays were performed in technical duplicates and at least in two independent assays.

## Assessment of the detection method combining ribozyme activity with HCR-FRET assays

After verifying the specific recognition of the SARS-CoV-2 viral RNA and optimizing the HCR-FRET measurement parameters, the detection method, which combines the specific cleavage of the SARS-CoV-2 RNA target by ribozymes and HCR-FRET assays, was implemented. Initially, three ribozymes were designed to cleave the consensus conserved 400 bp SARS-CoV-2 RNA target sequence identified from the MAFFT alignment (https://figshare. com/account/articles/25922725). The hammerhead ribozymes designed with RiboSoft and selected according to cleavage site, the specificity score, and the structure and accessibility scores are shown in Table 2. Ribozyme 2 had the highest accessibility and structure scores. However, ribozymes 1 and 3 had the highest specificity score, which is the most important parameter, as it is related to the possibility of off-targeting human RNA, according to the established parameters set. It is noteworthy that the designed ribozymes must be specific to SARS-CoV-2 RNA and not to human RNA present in most patient samples.

The ribozymes and the 400 nt target RNA were *in vitro* transcribed and analyzed by 2% agarose gel electrophoresis (S2 Fig). These results indicate that the transcription was successful and that the ribozymes were obtained in high concentration, showing the expected size. It is important to mention that the concentration of ribozyme 2 was lower than the concentration

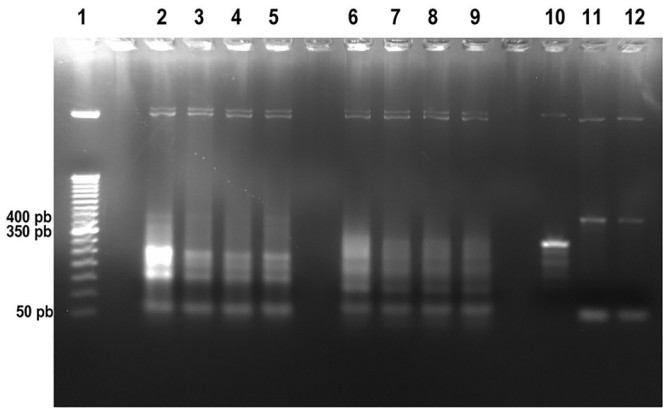

**Fig 5. Agarose gel electrophoresis of ribozyme catalytic assay products.** Ribozyme catalytic assay showing the RNA fragments obtained after ribozyme 1 (lanes 2–5) and ribozyme 3 (lanes 6–9) cleavage of the 400 nt target RNA molecule analyzed in 2% agarose gel electrophoresis. The target RNA is approximately 400 nt (lane 10), and the ribozyme 1 (lane 11) and ribozyme 3 (lane 12) of approximately 60 nt. RNA cleaved products for ribozyme 1 was 155 nt and 245 nt and for ribozyme 3 was 121 nt and 279 nt. Molecular Marker (MM) (lane 1) was the 50 bp DNA Ladder (Invitrogen).

of ribozymes 1 and 3. Subsequently, the catalytic activities of these ribozymes were assayed with the 400 pb target RNA in a 1:10 ratio (Fig 5). The results demonstrate that the ribozymes 1 and 3 showed cleavage activity resulting in the expected fragment sizes: 155 nt and 245 nt for ribozyme 1, and 121 nt and 279 nt for the ribozyme 3. Considering that ribozyme 2 was transcribed at a lower concentration (S2 Fig) and its cleavage activity assays resulted in several unexpected RNA fragments (S3 Fig), this ribozyme was not used in subsequent assays.

The Fig 6 shows the HCR-FRET process involving 1200 nM of 400 nt target RNA molecule without cleavage by ribozyme (Fig 6A) and cleaved separately by ribozymes 1 (Fig 6B) or 3 (Fig 6C). Experiments were performed with 600 nM H1 and H2 of first DNA hairpin set in SSC 5X buffer for a target RNA:H1:H2 (2:1:1) molar ratio performed in technical duplicates in three independent experiments. The schematic representations in the inset panels illustrate the mechanism of cleavage of RNA by ribozymes and the hybridization of H1/H2 with RNA in each experiment. Positive controls with the first 18 nt SARS-CoV-2 viral RNA target were used. The results indicate that the HCR-FRET process was unsuccessful with the intact 400 nt target RNA (Fig 6A), probably due to the difficulty of first set H1/H2 to perform HCR with large RNA (Fig 7A). However, successful HCR-FRET was observed in positive control using a smaller first 18 nt SARS-CoV-2 viral RNA target fragment as an initiator (Fig 6A). On the other hand, when the 400 nt RNA was cleaved separately by ribozymes 1 (Fig 6B) or 3 (Fig 6C), the HCR-FRET process was also unsuccessful using the first DNA hairpins H1/H2 set. However, positive controls using the smaller 18 nt fragment of the initiator RNA first target again demonstrated successful HCR-FRET process (Fig 6B and 6C). The results indicate that ribozyme cleavage activities are not followed by HCR-FRET onto the 245 nt (Fig 7B) and 279 nt (Fig 7C) fragments, using the first DNA hairpins H1 and H2 set.

Given the absence of HCR-FRET reactions in both digested and undigested RNA fragments by ribozymes (Fig 6B and 6C), a second set of DNA hairpins, H1 and H2, was designed using the NUPACK software to target another fragment produced by ribozyme activity. The potential to produce HCR-FRET with these two sets of DNA hairpins, H1/H2, was analyzed both individually and in combination (Fig 8A). The experiments used a 400 nt SARS-CoV-2 viral RNA segment as the target, with the HCR initiation sites for both sets of H1/H2 DNA hairpins

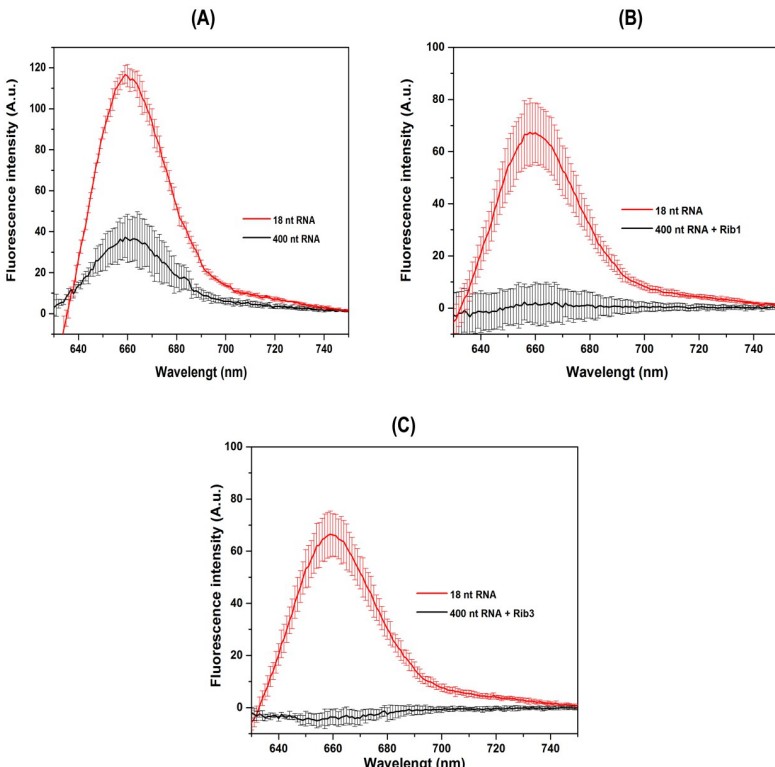

**Fig 6. Average fluorescence intensity emission spectra from HCR-FRET of RNA target cleaved by ribozymes using the first set of DNA hairpins, H1 and H2.** HCR-FRET assay with intact 400 nt target RNA (A), RNA cleaved with ribozyme 1 (B), and RNA cleaved with ribozyme 3 (C). Spectra were obtained by subtracting the negative control spectrum (without initiator RNA molecules) and the assays were performed with 1200 nM of RNA and 600 nM of H1 and H2 from first set in SSC 5X buffer for a target RNA:H1:H2 (2:1:1) molar ratio in technical duplicates in three independent days. In all panels, the red lines represent the positive control performed with the first 18 nt SARS-CoV-2 viral RNA target, and the black lines represent the intact 400 nt target RNA (A), RNA cleaved with ribozyme 1 (B), and RNA cleaved with ribozyme 3 (C).

are shown in Fig 9B. For the positive controls, the corresponding 18 nt SARS-CoV-2 viral RNA1 and RNA2 fragments were used for the first and second sets of H1/H2 hairpins, respectively. As expected, and consistent with the results shown in Fig 6A, the first set of H1/H2 DNA hairpins only successfully performed HCR-FRET with the 18 nt SARS-CoV-2 viral RNA1 segment used as a positive control (Fig 8A). However, the second set of designed H1/H2 DNA hairpins was able to perform HCR-FRET with both the 18 nt RNA2 target and the intact 400 nt RNA segment (Fig 8A). This unexpected result indicates that the second set of DNA hairpins can effectively perform HCR-FRET with both large (400 nt) and small (18 nt) RNA targets with equal efficiency.

When the experiment was conducted by mixing both sets of DNA hairpins, H1/H2, and using the two 18 nt SARS-CoV-2 viral RNA target positive controls, RNA1 and RNA2, a significant increase in the HCR-FRET signal was observed (Fig 8A). This increase was due to the simultaneous HCR-FRET in two different RNA1 and RNA2 target segments. Conversely, when the large 400 nt SARS-CoV-2 viral RNA target was used, HCR-FRET was like that observed with the second H1/H2 set using both the 18 nt and 400 nt RNA targets (Fig 8A). This result is expected, considering that only the second, and not the first, H1/H2 set can perform HCR-FRET with the large 400 nt RNA molecule, as previously demonstrated (Fig 6A).

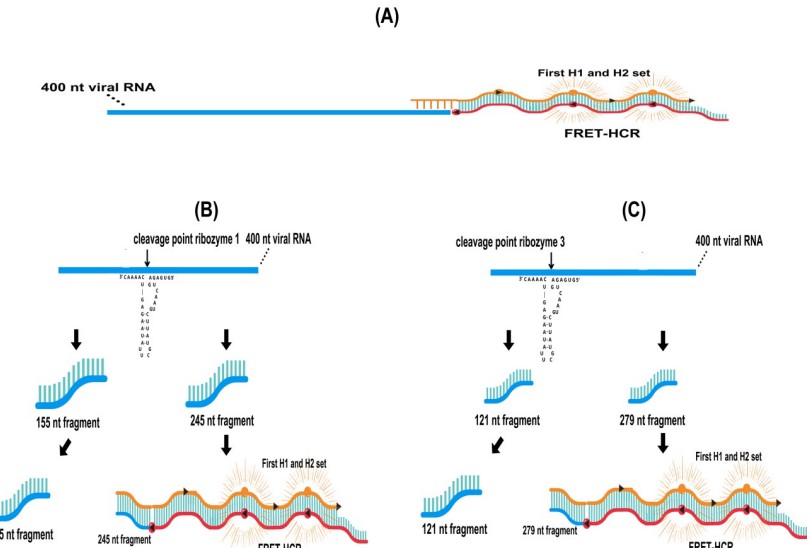

**Fig 7. Schematic representation of HCR-FRET by first DNA hairpins H1/H2 set and ribozymes cleavage.**
HCR-FRET sites of the first DNA hairpins H1/H2 set on intact 400 nt SARS-CoV-2 viral RNA (A). Ribozyme 1 cleaves the 400 nt SARS-CoV-2 viral RNA at one site, producing two fragments of 155 nt and 245 nt (B), while a single cleavage by ribozyme 3 generates the segments of 121 nt and 279 nt (C). HCR-FRET may occur in segments of 245 nt and 279 nt by the first set H1/H2 DNA hairpins. The cleavage sites are represented in order and scale position relative to each other.

These results indicate that only the second set of DNA hairpins, H1/H2, can perform HCR-FRET on both short and large fragments of viral RNA. In contrast, the first set of H1/H2 DNA hairpins is limited to small RNA molecules such as 18 nt.

To evaluate whether smaller RNA fragments could enhance the HCR-FRET process, aiming to improve diagnostic detection, experiments using ribozymes 1 and 3 separately, and in

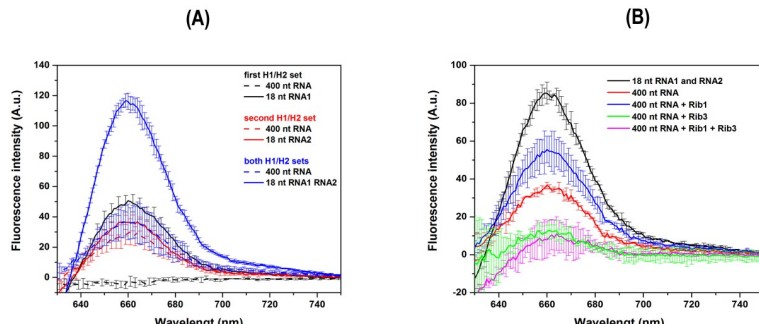

**Fig 8. HCR-FRET analyses using both H1/H2 DNA hairpins sets and ribozymes.** Average fluorescence intensity emission spectra resulting from HCR-FRET with each individual DNA hairpin set, the first H1/H2 set, the second H1/H2 set, and their combination (A). The experiments were conducted using 1200 nM of 400 nt SARS-CoV-2 viral RNA target and each 18 nt SARS-CoV-2 viral RNA segment, with 600 nM of each H1/H2 DNA hairpin, maintaining a molar ratio of 2:1:1 (target:H1:H2) in 5X SSC buffer. All assays were performed in biological duplicates across at least three independent experiments. Black lines represent the first H1/H2 set, red lines the second H1/H2 set, and blue lines the combination of both H1/H2 sets. Positive controls were conducted using the corresponding 18 nt initiator RNA molecules or both. Spectra were obtained by subtracting the negative control spectrum (without initiator RNA molecules). HCR-FRET assays were similarly performed but with prior cleavage of the 400 nt SARS-CoV-2 viral RNA by ribozymes 1 and 3, separately and both (B).

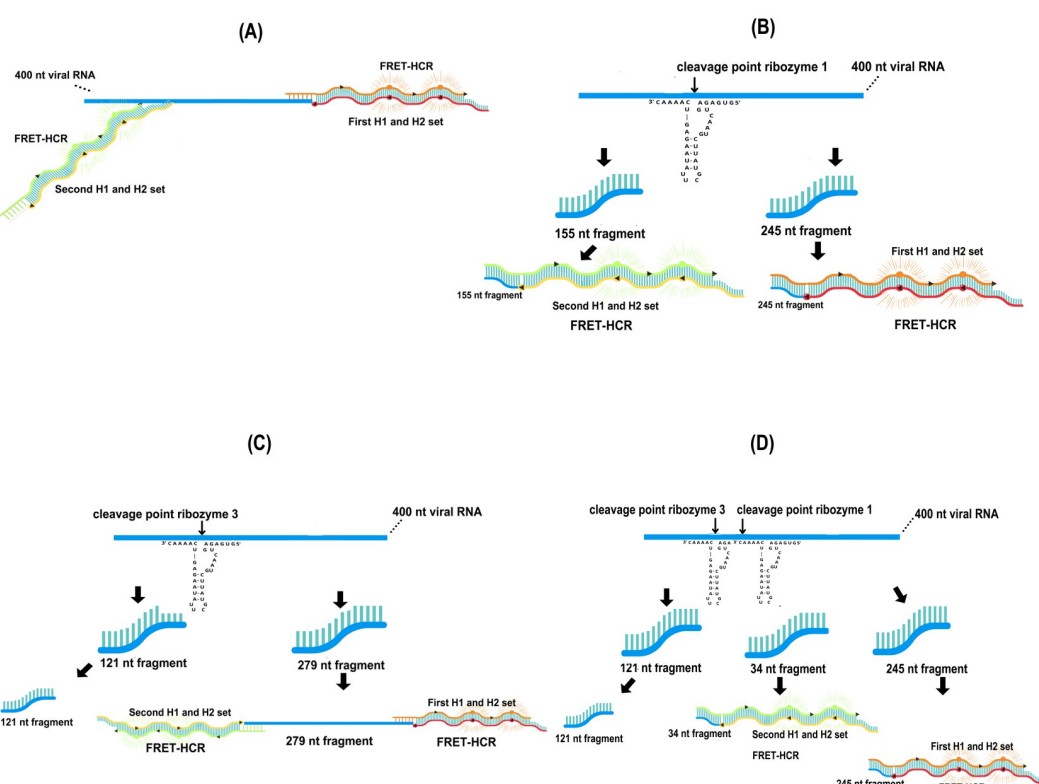

**Fig 9. Schematic representation of HCR-FRET by both H1/H2 DNA hairpins sets and ribozymes cleavage.** HCR-FRET sites of the first and second H1/H2 sets on intact 400 nt SARS-CoV-2 viral RNA (A). Ribozyme 1 cleaves the 400 nt SARS-CoV-2 viral RNA at one site, producing two fragments of 155 nt and 245 nt (B), while a single cleavage by ribozyme 3 generates the segments of 121 nt and 279 nt (C). HCR-FRET may occur on these segments by DNA hairpins H1/H2 sets. Simultaneous cleavage by ribozymes 1 and 3 produces three segments of 121 nt, 34 nt, and 245 nt, and HCR-FRET may occur in two of them (D). Cleavage sites are represented in order and scale position relative to each other.

combination, for digestion of the 400 nt RNA target were conducted (Fig 9B–9D). The positive controls were performed with both DNA hairpins H1/H2 sets and both 18 nt RNA initiator molecules (Table 1). The negative controls were prepared without RNA initiator molecules. The results showed an increase in HCR-FRET when the intact 400 nt RNA was digested with ribozyme 1 (Fig 8B), in agreement with the previous results showing the HCR-FRET of the first H1/H2 set with smaller RNA molecules (Fig 8A). According to the proposed HCR-FRET mechanism (Fig 9A and 9B) and the results shown (Fig 8A), the second H1/H2 DNA hairpins set can perform HCR-FRET on the intact 400 nt SARS-CoV-2 viral RNA target. Consequently, digestion of this viral RNA target with ribozyme 1 could potentially maintain and enhance the ability of the second H1/H2 set to perform HCR-FRET with the 155 nt as well as the first H1/H2 set with small RNA fragment of 245 nt (Fig 9B).

In contrast, digesting of the 400 nt SARS-CoV-2 viral RNA target with ribozyme 3 resulted in decreased HCR-FRET compared to the intact 400 nt RNA (Fig 8B). This digestion produced a 279 nt RNA fragment, which could be too large for HCR-FRET of the first H1/H2 set (Fig 9C). Furthermore, ribozyme 3 cleaves near the HCR-FRET initiation site of the second H1/H2 set, potentially generating a smaller fragment (Fig 9C) that could theoretically enhance HCR-FRET. However, the results contradict this hypothesis, as digestion with ribozyme 3 led to a significant decrease in HCR-FRET levels (Fig 8B). These findings may be due to ribozyme 3 interference with the recognition of RNA by the H1 DNA hairpin of the second set,

including potential cleavage or the formation of secondary structures of the 121 nt RNA fragment where H1 recognition should occur (Fig 9C) thus affecting HCR-FRET. Similar effects were observed with digestion using ribozymes 1 and 3 (Fig 8B), resulting in a decrease in HCR-FRET as observed with digestion by ribozyme 3. Additionally, the enhanced HCR-FRET previously observed with only ribozyme 1 digestion was completely lost. This unexpected result probably arises from the direct interference of ribozyme 3 or its digestion products in the HCR-FRET system. Another possibility is nonspecific cleavage by ribozyme 3 or the formation of secondary structures within or between the generated RNA fragments (Fig 9D), hindering the HCR-FRET.

One of the main limitations of methodologies based on thermo-sensitive enzymes is the need for refrigeration during transport and storage. To mitigate this requirement, ribozymes were dried using a speedvac to evaluate their stability without refrigeration. As shown in S4A Fig, the dried ribozymes maintained their cleavage activity, producing the expected fragment sizes: 155 nt for ribozyme 1 and 121 nt for ribozyme 3. These results indicate that the stability of the ribozymes remains unchanged whether they are dried, suggesting that the proposed method can be transported at any temperature. One of the main objectives in developing this methodology was to achieve a rapid diagnostic process, relative to other methods, by reducing the ribozyme RNA cleavage reaction time from 2 hours and 30 minutes to just 10 minutes, maintaining the effectiveness of the test. As shown in S4A Fig, ribozyme activity remains up to 10 min, suggesting that the total reaction time can be effectively reduced. Additionally, to establish the limit of detection, a series of assays were conducted to determine the lowest concentration at which the target viral RNA could be detected. The target concentrations tested were 0.5, 2.5, 5, 10, 50, 100, 300, and 600 nM. The limit of detection with two DNA hairpins H1/H2 sets was found to be around 50–100 nM (Fig 10A). To further improve the time efficiency of this method, a 10-minute ribozyme 1 cleavage assay was performed, immediately (0

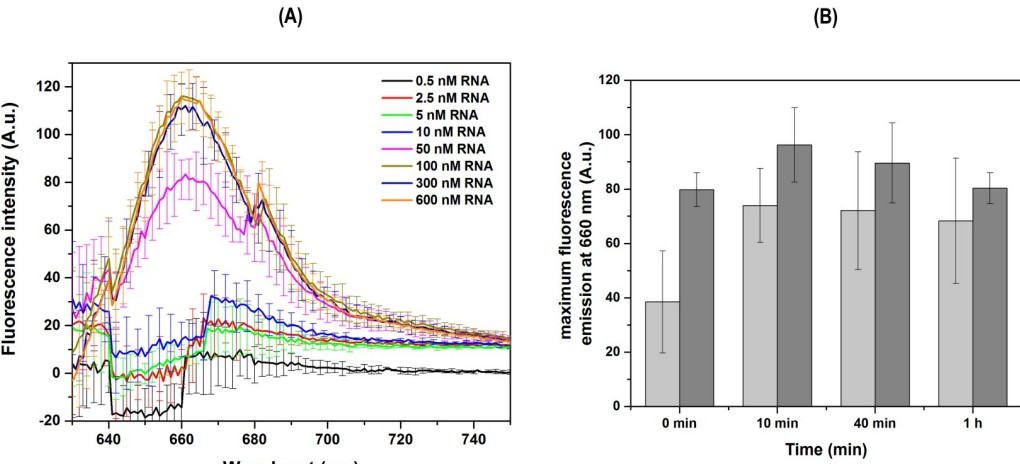

**Fig 10. Determination of the lowest detectable concentration of SARS-CoV-2 viral RNA target and detection time.** Average fluorescence intensity emission spectra from HCR-FRET between H1 and H2 at different concentrations of the 18 nt initiator RNA molecules (0.5, 2.5, 5, 10, 50, 100, 300, and 600 nM) and 600 nM of each H1/H2 set in 5X SSC buffer (A). Bar graph (B) showing the average fluorescence intensity emission at 660 nm after a 10-minute ribozyme 1 cleavage assay on intact 400 nt SARS-CoV-2 RNA viral, followed by fluorescence HCR-FRET detection at different time points: immediately (0 minutes), 10 minutes, 40 minutes, and one hour. The light gray bars represent cleave of 400 nt with ribozime 1 and dark gray bars the positive control. The positive control included both 18 nt initiator RNA1 and RNA2 targets at 600 nM and 600 nM of both DNA hairpins H1/H2 sets. All assays were performed in technical triplicates and at least two independent experiments. Spectra were obtained by subtracting the negative control spectrum (without initiator RNA molecules).

minutes), 10 minutes, 40 minutes, and one hour followed by spectrofluorometric analysis. The results showed the expected fluorescence intensity emission after 10 minutes (Fig 10B and S5B–S5D Fig), with an increase to approximately like the positive control signals.

## Manufacture of device

RGB LED modules (S6A and S6B Fig) were designed to generate a narrow range of wavelength excitations, which users can select using a keyboard-LCD display (S6C Fig). This design allows cost-effective use and high adaptability to excite initial donor fluorophores. This module is electronically controlled to regulate LED coloration and brightness. Wavelength calibration on the spectrofluorometer showed three light outputs spanning specific wavelengths and intensities, comprehending red (596–650 nm), green (490–560 nm), and blue (425–464 nm), with maximum wavelength of 632 nm, 514 nm, and 442 nm, respectively (S6D Fig). For controlling the modules, a circuit using the Arduino Nano microcontroller, TIP122 MOSFET transistors and 220 Ω resistors was designed for controlling the red, green, and blue channels (S7A and S7B Fig). The PWM pins of the Arduino board were used to control the different LEDs (S7C Fig). Arduino source code (version 1.8.9) makes use of the *analogWrite* function to implement PWM for each of the RGB LED channels (S1 Appendix). PWM values range from 0 (indicating the device is off) to 255 (representing the maximum voltage or power output), which allows to vary the output signal to create a range of colors by controlling the duty cycle of the PWM signal. A dark chamber or photographic module was specifically designed for digital image collection (S8A Fig), and the complete assembly of all device components is depicted in the final configuration (S8B and S8C Fig).

The FRET reactions (n = 5 technical replicates in two independent experiments) were prepared, each with different target RNA concentration (50, 100, 200, 300, 600, and 1200 nM) and the negative control as no target RNA. FRET verification in each reaction, was carried out on the SpectraMax M3 multimode microplate reader (S9 Fig). The results showed higher fluorescence intensities in the samples containing RNA target compared to the negative control (without target RNA) (Fig 11A). Interestingly, samples with RNA target concentrations of 100 nM or higher exhibited comparable intensity in the emission spectra, suggesting that the detection range extends to below 50 nM of RNA, indicating a low and sensitive RNA detection limit. These findings suggest FRET through a molecular recognition of the RNA target, homologous to viral RNA, by the fluorescent DNA-harpin molecules.

After demonstrating the FRET of the samples, the detection process was carried out using digital imaging. The samples were excited at an angle of 45˚ relative to the base of the plate and images were captured perpendicular to this base. Fully white plates were used for samples and positioned inside the dark chamber of the device for image capture using a Canon DS126191 camera. Digital photos were acquired using each of the three RGB LEDs (S6 Fig). When excited with red light (596–650 nm), a highly saturated intensity is observed in the red channel of the digital photo of the wells, possibly due to scattering of the red excitation light. The complete saturation of this channel's intensity made it difficult to process the images in the wells, resulting in absorbance values corrected by the software being very close to zero (Fig 11B). When the system is excited with blue light, a small increase in emission is observed in the red channel in all samples. As expected, no differences were found in the signals obtained for the samples and negative controls analyzed (Fig 11C). These results indicate that blue light is unable to excite the H1 fluorophore and does not result in FRET with H2. On the other hand, when digital photos were acquired under H1 excitation wavelength (λmax 547 nm) using a green LED (490–560 nm) an increased emission in the red channel was observed in wells photographs (Fig 11D). The results demonstrate a proportional increase in fluorescence emitted in

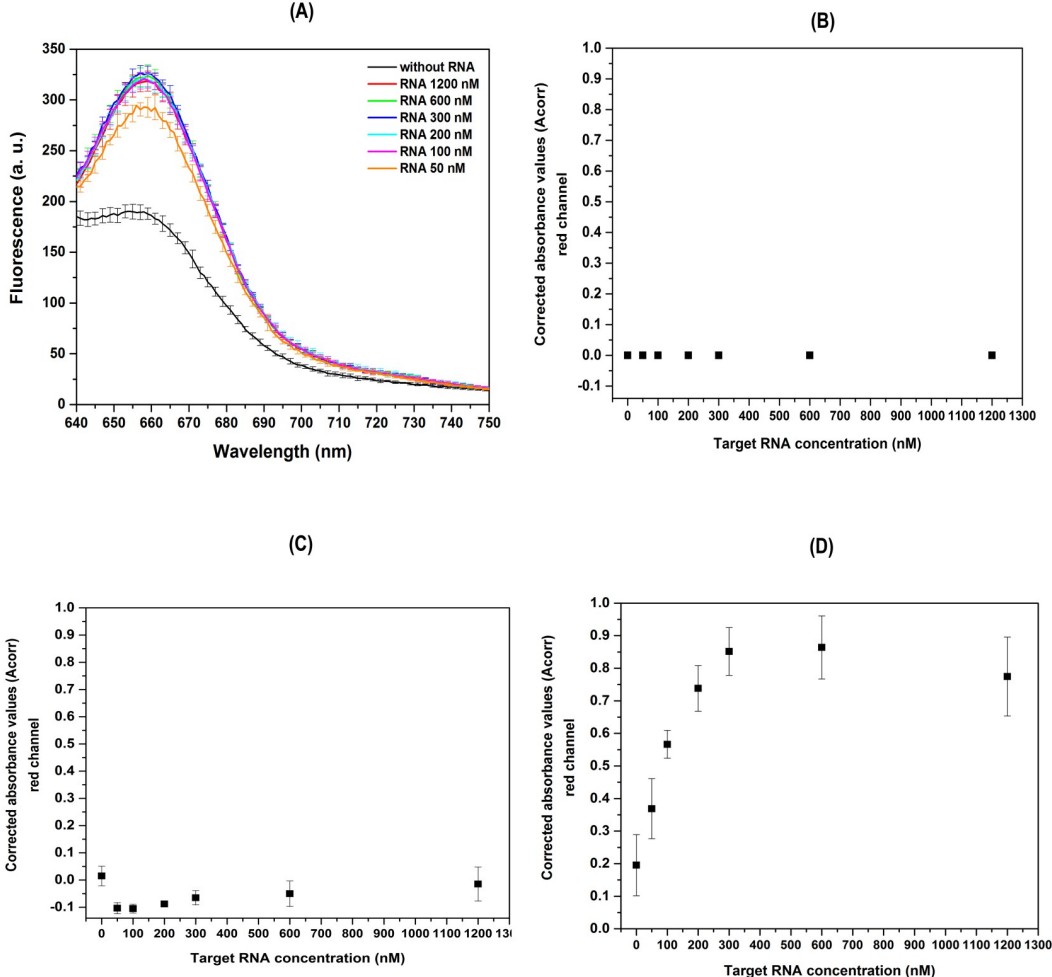

**Fig 11. Detection of the FRET process using prototype DIY photo-fluorometer based on image analysis.** Fluorescence intensity emitted from HCR-FRET with both H1 to H2 sets for different RNA target concentration (50, 100, 200, 300, 600, 1200 nM) (A). The readings were carried out using the SpectraMax M3 plate reader spectrofluorometer. FRET detection (red emission channel) was performed through image analysis using a Canon DS126191 digital camera with excitation by RGB LED light sources. The white plate sample excitation was performed at an incidence angle 45° relative to the base of the plate, considering that images would be captured perpendicular to this base. The results of excitation with red, blue, and green light are shown in panels (B), (C), (D), respectively.

the red channel between 100–300 nM of RNA target concentration. Concentrations below 50 nM are indistinguishable from the negative control, and above 300 nM, the red channel emission becomes saturated with increasing RNA concentration (Fig 11D).

## Discussion

Diagnostic methods based on nucleic acid amplification reactions, such as polymerase chain reaction (PCR), quantitative reverse transcription polymerase chain reaction (RT-qPCR), loop-mediated isothermal amplification (LAMP), helicase-dependent amplification (HDA), strand displacement amplification (SDA), and rolling circle amplification (RCA), are highly specific and sensitive. Several of these methods are used as viral diagnostic tests. However, other methodologies, such as those based on hybridization chain reaction (HCR), are potential

cost-effectiveness alternatives compared to conventional thermal amplification reactions of nucleic acid. HCR can be performed under less parameterized conditions, without the need for more complex thermal cycling processes and the use of high precision instruments, thus reducing experimental costs [36–39].

Additionally, HCR exhibits high sensitivity and specificity, with the capability to recognize single-base mismatched DNA [40, 41]. Moreover, the simplicity of HCR allows it to be integrated into various diagnostic platforms, including point-of-care devices, making it a versatile tool in molecular diagnostics. The ability to be conducted at a constant temperature further enhances its application in regions with limited sources, where traditional thermal cycling equipment is not feasible [42].

In this study, the bases for an innovative RNA detection methodology were introduced using SARS-CoV-2 as a proof of concept. Our approach involves the concurrent utilization of a trans-acting extended hammerhead ribozyme and Cy3/Cy5-labeled metastable DNA hairpins in Fluorescence Resonance Energy Transfer-Hybridization Chain Reaction (HCR-FRET). This pioneering method operates enzymatically autonomously, eliminating the need for protein enzyme-based amplification processes. Notably, it offers distinct advantages over conventional tests, including the removal of refrigeration requirements by incorporating stable, lyophilizable nucleic acids molecules instead of protein enzymes. Furthermore, our approach introduces an innovative aspect: the detection of the viral RNA fragment, produced through ribozyme cleavage, driving to HCR-FRET via hybridization DNA hairpins, enabling rapid signal detection within 20–30 minutes of total time with a detection limit of 100 nM of RNA target, without the need for protein enzymatic steps, a feature inherent in existing methods.

Addressing the historical context of COVID-19 diagnostics, the gold standard method, RT-qPCR, initially faced limitations, particularly in the laborious and costly RNA extraction step. To overcome this disadvantage, Smyrlaki and collaborators simplified the test by identifying SARS-CoV-2 by RT-qPCR directly on heat-inactivated samples [32]. However, this methodology still presents some other limitations regarding the need for refrigeration, time-consuming, and the requirement of highly specialized equipment and labor. In contrast, the proposed methodology for viral RNA identification, using SARS-CoV-2 as proof of concept, employs specific ribozymes, simple molecules, eschewing the need for enzymes and specialized equipment. The results demonstrate that the proposed method not only requires less specialized equipment and labor but also achieves 100 nM detection sensitivity, as demonstrated through HCR-FRET assays employing Varioskan LUX and SpectraMax M3 spectrofluorometers (Figs 3 and 10A). Furthermore, the fluorescence signal could be detected in approximately 30–40 minutes (S4 Fig and Fig 10B) instead of approximately hours taken by RT-qPCR [43–45].

This work distinguishes itself from previous studies by integrating ribozymes with HCR-FRET, providing a novel approach to diagnosing SARS-CoV-2 RNA and other viruses. Unlike conventional HCR assays that require electrophoresis gel analysis, our methodology relies on detecting fluorescence emission, eliminating the requirement for skilled labor and specialized equipment [46]. The fluorescence intensity observed in the experiments, along with the potential for signal captures using a prototype DIY photo-fluorometer device incorporating a digital camera (S9 Fig and Fig 11), and highlights the practicality and accessibility of the proposed diagnostic approach. Detection of fluorescence emission, facilitated by digital cameras, aligns with our envisioned objectives. The fluorescence intensity signals generated by the described detection method, utilizing the 10 minutes ribozyme activity assay followed by 10–40 minutes HCR-FRET reactions, exceed background levels, underscoring its potential for rapid and user-friendly diagnosis. This not only enhances test accessibility and portability but also significantly reduces diagnostic complexity. Moreover, the ease of sharing captured images for remote interpretation and analysis holds promise for resource-limited settings.

The utilization of digital image analysis has been widely applied across diverse scientific domains, notably in chemistry [47–51]. This approach offers distinct advantages, including precision and user-friendly operation, and has demonstrated substantial value in numerous scientific applications. Within this approach, the integration of digital camera with image-based detection methods represents a significant leap forward in conducting fluorescence tests with enhanced accessibility and efficiency. Fluorescence spectroscopy is the most widely used optical spectroscopic method in analytical chemistry and scientific research. However, a standard spectrofluorometer is expensive and comprises several essential components: a light source, wavelength selector, sample cell, a secondary wavelength selector to isolate emitted fluorescence, and the detector. Notably, integrating digital image analysis into compact devices for fluorescence detection using a digital camera significantly reduces the cost of diagnosis. In digital images, although the distinction between positive and negative samples may not be visually apparent to the naked eye, it becomes significantly clearer when pixels are analyzed for RGB values using specialized software, showing a difference of almost an order of magnitude.

One of the reasons for this behavior could be due to the human eyes is imperfect in their ability to distinguish between the three primary colors. For example, both red and blue cone cells in the human eye respond to wavelengths around 420 nm, resulting in a violet color that can appear very similar to purple, which is a mixture of red and blue colors [52]. On the other hand, digital cameras are equipped with advanced image sensors such as CCD (Charge-Coupled Device) and CMOS (Complementary Metal-Oxide-Semiconductor). These sensors capturing light through photodiodes (or pixels), converting it into electrical signals. While these sensors start as monochromatic, they utilize color filters positioned in front of the pixels to selectively allow specific wavelengths of light to pass through. Commonly used RGB (Red, Green, Blue) filters enable the camera to capture detailed color information from the scene [53]. Therefore, utilizing a DIY photo-fluorometer device with a digital camera could enhance practicality and accessibility for diagnostic approaches involving RNA/DNA viruses.

## Conclusion

Considering future pandemics such as COVID-19, it is a necessary to develop simple, cost-effective, user-friendly, accessible, and quickly adaptable diagnostic methods for new viruses and emerging pathogens. In this context, the development of diagnostic methods based on HCR-FRET can result in high-throughput initial screening. In the case of COVID-19, even after the World Health Organization (WHO) declared the end of the COVID-19 public health emergency, the emergence of the EG.5 variant of SARS-CoV-2 led to new outbreaks, highlighting the importance of developing new technologies [54]. It is important to emphasize that the proposed methodology is not intended to replace highly specific and sensitive methods based on enzymatic amplification thermocycles, such as PCR and RT-qPCR. Rather, it can serve as a foundation for alternative viral diagnostic methods that address resource limitations and the need for specialized expertise and equipment.

In this study, we developed a novel molecular method for detecting viral genome fragments. As a proof of concept, we used SARS-CoV-2 RNA viral fragments. The method is based on trans-acting ribozymes and FRET-based hybridization of fluorescent DNA hairpins. This method is flexible and easily adaptable for use with other viruses. Rational designs were successful in producing at least two active ribozymes and two sets of H1 and H2 DNA reporter hairpin molecules. Fluorescence assays conducted on different spectrophotometers detected HCR-FRET amplification via hybridization of fluorescent DNA hairpins in DNA that mimicking viral RNA, and in RNA fragments generated by ribozymes. The design and construction of a DIY photo-fluorometer prototype allowed to explore the development of a simple and cost-

effective point-of-care detection method based on digital image analysis. Future efforts should focus on refining critical parameters, such as enhancing method sensitivity through multiplexing H1 and H2 molecules, optimizing ribozymes, and conducting a comprehensive exploration of the method's minimum detection threshold to ensure robust validation.

## Supporting information

**S1 Appendix. The Arduino sketch utilizes I2C communication to interface with a keypad and an LCD display.** The system allows users to select a color (red, green, or blue) in external component RGB LED using the keypad. The selected color is then displayed on corresponding RGB LED with feedback provided through the buzzer and visual confirmation on the LEDs and LCD. The code includes several libraries such as I2C communication with devices (Wire. h), supports I2C communication for the keypad (Keypad_I2C.h), managing the keypad input (Keypad.h), and handling interactions with the I2C-connected LCD display (LiquidCrystal_I2C.h).
(DOCX)

**S1 Fig. Absorption and fluorescence spectra of H1 and H2.** Absorption spectra of H1-Cy3 and H2-Cy5 (black lines), along with their corresponding fluorescence emission spectra (red lines) (A). H1 and H2 were excited at their respective absorption maxima of 547 nm and 647 nm. The spectra were obtained using a SpectraMax M3 plate reader, with each spectrum representing the mean and standard deviation of three replicates.
(TIF)

**S2 Fig. Target RNA and ribozymes transcribed in vitro in 2% agarose gel electrophoresis.** The ribozyme 1 (lane 2), ribozyme 2 (lane 3) and ribozyme 3 (lane 4) are approximately 60 nt. The target RNA is approximately 400 nt (lane 5). Invitrogen 50 bp DNA Ladder molecular marker (lane 1).
(TIF)

**S3 Fig. Agarose gel electrophoresis of ribozymes catalytic assay.** Agarose gel electrophoresis displaying RNA fragments obtained after cleavage of a 400 nt SARS-CoV-2 viral RNA target by ribozymes 1, 2 and 3. Controls for free ribozymes 1, 2, 3, and the RNA target are shown in lanes 2, 3, 4 and 5, respectively. The cleavage products are shown for ribozyme 1 (lane 6), ribozyme 3 (lane 7), and ribozyme 2 (lane 8). The target RNA is approximately 400 nt, and the ribozymes are approximately 60 bp each. Cleavage products for ribozyme 1 are 155 nt and 245 nt (lane 6), while for ribozyme 3 they are 121 nt and 279 nt (lane 7). Cleavage with ribozyme 2 resulted in several unexpected RNA fragments (lane 8).
(TIF)

**S4 Fig. Agarose gel electrophoresis of dried ribozymes catalytic assay.** (A) Agarose gel electrophoresis displaying RNA fragments obtained after cleavage of a 400 nt SARS-CoV-2 viral RNA target by ribozyme 1 and/or ribozyme 3. The cleavage products are shown for ribozyme 1 (lanes 2–4), ribozyme 3 (lanes 5–7), and both ribozymes combined (lanes 8–10). (B) Evaluation of cleavage activity at various time points (10, 30, 60, 90, and 150 minutes) with ribozyme 1 (lanes 2–6), ribozyme 3 (lanes 7–11), and both ribozymes combined (lanes 12–16). Controls for ribozyme 1, ribozyme 3, and the RNA target are shown in lanes 12, 13, and 14 (Panel A), and lanes 17, 18, and 19 (Panel B), respectively. The target RNA is approximately 400 nt, and the ribozymes are approximately 60 bp each. Cleavage products for ribozyme 1 are 155 nt and 245 nt, while for ribozyme 3 they are 121 nt and 279 nt. The molecular marker (MM) used is

the 50 bp DNA Ladder (Invitrogen), shown in lane 1 in both panels.
(TIF)

**S5 Fig. Evaluation of measurement time for detecting HCR-FRET on SARS-CoV-2 RNA viral fragment.** Average fluorescence intensity emission spectra from HCR-FRET between 600 nM of each DNA hairpin (H1 and H2) in 5X SSC buffer at different detection times: immediately (0 minutes), 10 minutes, 40 minutes, and one hour. Cleavage of intact 400 nt SARS-CoV-2 RNA viral fragment by ribozyme 1 was performed during 10 min. The positive control included both 18 nt initiator RNA1 and RNA2 targets at 600 nM and 600 nM of both DNA hairpins (H1 and H2). All assays were performed in technical triplicates and at least two independent experiments. Spectra were obtained by subtracting the negative control spectrum (without initiator RNA molecules).
(TIF)

**S6 Fig. Device features of the RGB LED modules integrated into a 96-well plate reader for a DIY photo-fluorimeter prototype based on image analysis.** The RGB LED modules were developed and manufactured as a compact module (A) with diffuser plate (spreading board) to scatter light effectively (B). The LED modules are electronically controlled by control module through an Arduino nano V3 Atmega328 Ch340 microcontroller board, employing three TIP122 MOSFET transistors assisted by 220 Ω resistors (C). Wavelength calibration on the spectrofluorometer showed three light output spanning specific wavelengths and intensities, comprehending red (596–650 nm), green (490–560 nm), and blue (425–464 nm), with maximum wavelength of 632 nm, 514 nm, and 442 nm, respectively (D). The plastic structural components were designed using CAD software Dassault Systems SolidWorks and manufactured using 3D printing.
(TIF)

**S7 Fig. Control module of the prototype DIY photo-fluorimeter based on image analysis.** The electronic control module include an user interface module with a 2x16 LCD display with i2c module and 3x4 matrix keyboard and passive buzzer (A), a 12V DC connector plug, On/Off switch, three TIP122 MOSFET transistors assisted by 220 Ω resistors, an Arduino nano V3 Atmega328 Ch340 microcontroller board, LM2596 voltage regulator, a PCF8574 Digital I/O Expander via I2C-bus by serial clock (SCL), serial data (SDA) (B), and the control software. Electronic diagram of the control circuit using the Arduino Nano microcontroller is shown in (C). The plastic structural components were designed using CAD software Dassault Systems SolidWorks and manufactured using 3D printing.
(TIF)

**S8 Fig. Dark chamber and final assembly of the prototype DIY photo-fluorimeter based on image analysis.** The dark chamber or photographic module designed specifically for collecting digital images (A), and its assembly on the control module (B). Complete assembly of all components or modules of the prototype DIY photo-fluorimeter based on image analysis, showing the fully operational device (C).
(TIF)

**S9 Fig. Strategy for FRET verification and detection using prototype DIY photo-fluorimeter based on image analysis.** FRET reactions were prepared on black flat bottom microplate and FRET verification in each reaction was carried out on the SpectraMax M3 multimode microplate reader. These samples were transferred to fully white microplate and positioned inside the dark chamber of the device for image capture using a Canon DS126191 camera. Digital photos were acquired using each of the three RGB LEDs detection process involved digital

imaging.
(TIF)

## Acknowledgments

We thank Roberto Togawa for his assistance with the computational alignment analysis.

## Author Contributions

**Conceptualization:** Aisel Valle Garay, Sonia Maria de Freitas, Cíntia Marques Coelho.

**Data curation:** Leonardo Ferreira da Silva, Aisel Valle Garay, Pedro Felipe Queiroz, Sophia Garcia de Resende, Mayna Gomide, Izadora Cristina Moreira de Oliveira, Amanda Souza Bernasol, Anibal Arce, Liem Canet Santos, Cíntia Marques Coelho.

**Formal analysis:** Leonardo Ferreira da Silva, Aisel Valle Garay, Pedro Felipe Queiroz, Sophia Garcia de Resende, Mayna Gomide, Izadora Cristina Moreira de Oliveira, Amanda Souza Bernasol, Anibal Arce, Liem Canet Santos, Sonia Maria de Freitas, Cíntia Marques Coelho.

**Funding acquisition:** Fernando Torres, Sonia Maria de Freitas, Cíntia Marques Coelho.

**Investigation:** Leonardo Ferreira da Silva, Aisel Valle Garay, Pedro Felipe Queiroz, Sophia Garcia de Resende, Mayna Gomide, Izadora Cristina Moreira de Oliveira, Amanda Souza Bernasol, Anibal Arce, Liem Canet Santos, Cíntia Marques Coelho.

**Methodology:** Leonardo Ferreira da Silva, Aisel Valle Garay, Mayna Gomide, Izadora Cristina Moreira de Oliveira, Sonia Maria de Freitas, Cíntia Marques Coelho.

**Project administration:** Sonia Maria de Freitas.

**Resources:** Fernando Torres, Ildinete Silva-Pereira, Sonia Maria de Freitas.

**Software:** Aisel Valle Garay, Mayna Gomide, Anibal Arce.

**Supervision:** Fernando Torres, Ildinete Silva-Pereira, Sonia Maria de Freitas, Cíntia Marques Coelho.

**Visualization:** Leonardo Ferreira da Silva, Aisel Valle Garay, Pedro Felipe Queiroz, Amanda Souza Bernasol, Anibal Arce.

**Writing – original draft:** Leonardo Ferreira da Silva, Aisel Valle Garay, Pedro Felipe Queiroz, Sophia Garcia de Resende, Mayna Gomide, Izadora Cristina Moreira de Oliveira, Amanda Souza Bernasol, Anibal Arce, Liem Canet Santos, Fernando Torres, Ildinete Silva-Pereira, Sonia Maria de Freitas, Cíntia Marques Coelho.

**Writing – review & editing:** Leonardo Ferreira da Silva, Aisel Valle Garay, Pedro Felipe Queiroz, Sophia Garcia de Resende, Mayna Gomide, Izadora Cristina Moreira de Oliveira, Amanda Souza Bernasol, Anibal Arce, Fernando Torres, Ildinete Silva-Pereira, Sonia Maria de Freitas, Cíntia Marques Coelho.

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
