## [Decision Letter · Decision Letter 0]

1 Jul 2024

PONE-D-24-22185A novel viral RNA detection method based on the combined use of trans-acting ribozymes and HCR-FRET analysesPLOS ONE

Dear Dr. Marques Coelho,

Thank you for submitting your manuscript to PLOS ONE. After careful consideration, we feel that it has merit but does not fully meet PLOS ONE’s publication criteria as it currently stands. Therefore, we invite you to submit a revised version of the manuscript that addresses the points raised during the review process.

We look forward to receiving your revised manuscript.

Kind regards,

Ruijie Deng

Academic Editor

PLOS ONE

Journal Requirements:

4. We note that your Data Availability Statement is currently as follows: All relevant data are within the manuscript and its Supporting Information files

7. PLOS ONE now requires that authors provide the original uncropped and unadjusted images underlying all blot or gel results reported in a submission’s figures or Supporting Information files. This policy and the journal’s other requirements for blot/gel reporting and figure preparation are described in detail at https://journals.plos.org/plosone/s/figures#loc-blot-and-gel-reporting-requirements and https://journals.plos.org/plosone/s/figures#loc-preparing-figures-from-image-files. When you submit your revised manuscript, please ensure that your figures adhere fully to these guidelines and provide the original underlying images for all blot or gel data reported in your submission. See the following link for instructions on providing the original image data: https://journals.plos.org/plosone/s/figures#loc-original-images-for-blots-and-gels.   

Reviewers' comments:

Reviewer's Responses to Questions

**Comments to the Author**

1. Is the manuscript technically sound, and do the data support the conclusions?

Reviewer #1: Partly

Reviewer #2: Partly

2. Has the statistical analysis been performed appropriately and rigorously? 

Reviewer #1: Yes

Reviewer #2: Yes

3. Have the authors made all data underlying the findings in their manuscript fully available?

Reviewer #1: Yes

Reviewer #2: Yes

4. Is the manuscript presented in an intelligible fashion and written in standard English?

Reviewer #1: Yes

Reviewer #2: Yes

5. Review Comments to the Author

Reviewer #1: In this manuscript, the authors developed an assay for detecting SARS CoV-2 RNA viral based on trans-acting ribozymes and FRET-based hybridization of fluorescent DNA hairpins. The target RNA is specifically recognized and cleaved by ribozymes, releasing an initiator fragment that triggers a hybridization chain reaction (HCR) with DNA hairpins containing fluorophores, leading to a FRET process. The manuscript has abundant experimental data, but remains some questions to be addressed.

1. Please simplify the abstract section. It is suggested not to include too much research background in the abstract.

2. The introduction section does not adequately demonstrate the advancements of the HCR-based assay. The authors should discuss the progress made in this area and highlight the key challenges.

3. The schematic figures are simple and need to be improved to meet the journal's standards.

4. The details of gel electrophoresis procedures should be provided.

5. This method utilizes the trans-acting ribozymes to recognize SARS-CoV-2 RNA. Is it promising to identify single nucleotide mutations in viral RNA?

6. The authors indicated that this method is cost-effective and user-friendly. Therefore, the detection cost and preprocessing requirements of this method need to be discussed.

Reviewer #2: The manuscript presents an innovative approach to viral RNA detection, focusing on the development of a method that combines trans-acting ribozymes with Hybridization Chain Reaction-Fluorescence Resonance Energy Transfer (HCR-FRET) analysis for the detection of SARS-CoV-2 viral RNA. The proposed method offers the advantages of speed, cost-effectiveness, ease of use, and does not require specialized equipment, making it particularly suitable for settings with limited resources. The research team has also designed and constructed a prototype for a Do-It-Yourself (DIY) photo-fluorometer, further exploring the potential for a simple and cost-effective point-of-care detection method based on digital image analysis. Overall, the study provides a novel molecular mechanism for the detection of viral RNA and shows promise for broad application I recommend publication of this article after addressing the following concerns.

1. Figure 1 mentioned the “Hammerhead Ribozyme”, but there is no introduction to the “Hammerhead”.

2. A comparison with existing diagnostic methods such as RT-qPCR is suggested to highlight the advantages of the proposed method in terms of sensitivity, specificity, and cost-effectiveness.

3. Some figures require improvement in clarity and labeling to ensure accurate conveyance of information.

4. SARS-CoV-2 detection in clinical throat swab samples should be investigated.

5. I recommend authors to go through another round of editing prior to submission of the revision.

6. PLOS authors have the option to publish the peer review history of their article (what does this mean?). If published, this will include your full peer review and any attached files.

Reviewer #1: No

Reviewer #2: No

---

## [Author Response · Author response to Decision Letter 0]

22 Jul 2024

Responses to the reviewer 1 comments: 

In this manuscript, the authors developed an assay for detecting SARS CoV-2 RNA viral based on trans-acting ribozymes and FRET-based hybridization of fluorescent DNA hairpins. The target RNA is specifically recognized and cleaved by ribozymes, releasing an initiator fragment that triggers a hybridization chain reaction (HCR) with DNA hairpins containing fluorophores, leading to a FRET process. The manuscript has abundant experimental data, but remains some questions to be addressed.

1. Please simplify the abstract section. It is suggested not to include too much research background in the abstract. 

2. The introduction section does not adequately demonstrate the advancements of the HCR-based assay. The authors should discuss the progress made in this area and highlight the key challenges. 

3. The schematic figures are simple and need to be improved to meet the journal's standards. 

4. The details of gel electrophoresis procedures should be provided. 

5. This method utilizes the trans-acting ribozymes to recognize SARS-CoV-2 RNA. Is it promising to identify single nucleotide mutations in viral RNA? 

6. The authors indicated that this method is cost-effective and user-friendly. Therefore, the detection cost and preprocessing requirements of this method need to be discussed. 

1. Please simplify the abstract section. It is suggested not to include too much research background in the abstract. 

We agree with the reviewer and have simplified the abstract section by removing the following text about background knowledge (lines 28-32):

These include the need for refrigeration, specialized equipment, and professional expertise. Therefore, there is a need for a diagnostic method that is rapid, cost-effective, user-friendly, and does not require specialized equipment compared to other techniques. Such a method would be extremely useful in resource-limited settings, enhancing readiness for future epidemics and pandemics.

2. The introduction section does not adequately demonstrate the advancements of the HCR-based assay. The authors should discuss the progress made in this area and highlight the key challenges.

We agree with the reviewer and have replaced the following text (Lines 78-88) in the Introduction section:

This methodology involves two hairpin DNA probes, each labeled with fluorophores, containing complementary regions that, upon encountering a target initiator, concatenate by hybridization to form an elongated double-stranded DNA structure. This enables the detection of the acid nucleic target via situ hybridization and through the enhanced amplification fluorescence signals (14-16). However, this HCR method requires a washing step to removal non-binding probes. To optimize this stage and broaden the methodology's applicability, a Fluorescence Resonance Energy Transfer (FRET) reaction has been incorporated into the HCR approach (17-19). The FRET operates based on the transference of energy fluorescence between donor-acceptor molecular pairs, making it a highly efficient method for diagnostic applications. It offers the advantage of single-step execution and specific identification of the target molecules using a fluorophore-labeled molecular probe and smartphones (20).

by:

The concept of HCR was introduced by Dirks and Pierce in 2004 (15). In HCR, stable DNA monomers are assembled only in the presence of a target DNA or RNA fragment. This process involves two stable DNA hairpins, containing complementary regions, which coexist in solution until the initiator strands trigger a hybridization cascade, forming nicked double helices. The molecular weight of HCR products varies inversely with initiator concentration, allowing DNA to act as an amplifying transducer for biosensing applications (15, 16). Despite its advantages, HCR faces challenges that need to be addressed. Optimizing reaction conditions for consistent performance, improving the speed of the HCR process, and developing robust protocols for multiplexed detection are key areas needing advancement.

To optimize it and broaden the methodology's applicability, a Fluorescence Resonance Energy Transfer (FRET) reaction has been incorporated into the HCR approach. The FRET operates based on the transference of energy fluorescence between donor-acceptor molecular pairs, making it a highly efficient method for diagnostic applications (16-18). This methodology HCR-FRET has been applied to identify and image mRNA in situ using fluorescence microscopy, and to detect tumor-related mRNA in single cells and tissue sections with high sensitivity (19, 20). This ability is promising for early cancer diagnosis by distinguishing between cancer and normal cells (16). Thus it, offers the advantage of single-step execution and specific identification of the target molecules using a fluorophore-labeled molecular probe and smartphones (20).

New references have been included and updated.

3. The schematic figures are simple and need to be improved to meet the journal's standards. 

Thank you for your valuable feedback regarding the schematic figures. In response, we have revised the figures to ensure they meet the journal's standards and comply with the guidelines for authors. The updated figures are now formatted as high-quality .TIFF files with LZW compression. They are in RGB color mode (8 bit/channel), with dimensions ranging from 789 to 2250 pixels in width (at 300 dpi) and a maximum height of 2625 pixels (at 300 dpi). The resolution of the figures are 600 dpi. All text within the figures is set in Arial font, ranging from 8 to 12 points in size. Additionally, multi-panel figures have been combined into a single file for clarity and ease of review.

4. The details of gel electrophoresis procedures should be provided. 

We appreciate the reviewer's suggestion and have added the electrophoresis procedure details to the Materials and Methods section (at lines 169):

Horizontal agarose gel electrophoresis was performed using RNase-free low melting agarose 2% (w/v) in a 15 x 15 cm UV-transparent tray system with PowerPac™ Basic Power Supply (Bio-Rad Laboratories, Inc., Hercules, California, USA). The electrophoresis was conducted with 1x Tris/Acetic Acid/EDTA (TAE) nucleic acid electrophoresis buffer solution at pH 8.3, at a constant 80 Volts. Subsequently, the gels were stained with 0.01 mg/mL of ethidium bromide (35).

Reference 39 was included.

5. This method utilizes the trans-acting ribozymes to recognize SARS-CoV-2 RNA. Is it promising to identify single nucleotide mutations in viral RNA? 

We appreciate the reviewer's question regarding the use of trans-acting ribozymes for the detection of SARS-CoV-2 RNA and their potential for identifying single nucleotide mutations in viral RNA.

To address this, our strategy for selecting target ribozyme sequences was designed to ensure specific identification of SARS-CoV-2. We focused on a 400 nt fragment previously used by other authors to develop the test for SARS-CoV-2 identification (Reference 30 in the first manuscript: Hou T, Zeng W, Yang M, Chen W, Ren L, Ai J, et al. Development and evaluation of a rapid CRISPR-based diagnostic for COVID-19. PLoS pathogens. 2020;16(8):e1008705). The fragment is within the ORF1ab region (YP_009724389.1, positions 11,459-11,858) corresponding to the non-structural protein 6 (Nsp6). Nsp6 is a highly conserved membrane protein with six transmembrane domains, essential for viral replication. Due to its crucial role, it exhibits high sequence conservation among SARS-CoV-2 strains.

In our study, we aligned this 400 nt fragment with 16,000 consensus sequences of SARS-CoV-2, achieving over 98% identity. This high level of conservation makes it an ideal target for ribozyme design. We selected three ribozymes targeting this region and evaluated their specificity by aligning their target sequences with genomes of related coronaviruses (SARS-CoV, MERS-CoV, and other common human coronaviruses).

The results showed significant differences in nucleotide sequences between SARS-CoV-2 and other coronaviruses, confirming the specificity of our selected ribozymes. For instance:

Ribozyme 1 (22 nts):

• Genome-229: 10 nucleotide differences

• Bat SARS-like genome: 5 nucleotide differences

• HKU1 genome: 12 nucleotide differences

• MERS control and MERS NC genomes: 9 nucleotide differences

• NL63 genome: 10 nucleotide differences

• OC43 genome: 10 nucleotide differences

• SARS Tor 2 genome: 8 nucleotide differences

Ribozyme 2 (30 nts):

• Genome-229: 13 nucleotide differences

• Bat SARS-like genome: 7 nucleotide differences

• HKU1 genome: 13 nucleotide differences (including 1 deletion and 1 insertion)

• MERS control and MERS NC genomes: 12 nucleotide differences

• NL63 genome: 12 nucleotide differences

• OC43 genome: 15 nucleotide differences (including 1 deletion and 1 insertion)

• SARS Tor 2 genome: 8 nucleotide differences

Ribozyme 3 (23 nts):

• Genome-229: 4 nucleotide differences

• Bat SARS-like genome: 2 nucleotide differences

• HKU1 genome: 7 nucleotide differences

• MERS control and MERS NC genomes: 6 nucleotide differences

• NL63 genome: 7 nucleotide differences

• OC43 genome: 7 nucleotide differences

• SARS Tor 2 genome: 2 nucleotide differences

These alignments suggested that our ribozymes are highly specific to SARS-CoV-2 and differ significantly from other related viruses. This specificity indicates that our ribozyme-based method could be promising for identifying single nucleotide mutations in SARS-CoV-2 RNA, enhancing its potential utility in detecting and monitoring viral mutations. However, experimental assays are necessary to confirm this hypothesis.

Additionally, to better clarify the analysis carried out, we replaced the following text of the manuscript (lines 313-318):

To confirm that the target sequence represents a consensus to the SARS-CoV-2 variants, genomic alignment analysis of different viral strains was performed and available (https://figshare.com/account/articles/25922725). The selected conserved sequence was, then, compared to sequences from other human viruses such as the SARS and MERS families, to withdraw the sequences that are also consensual for these other viruses.

by

The target sequence is a 400 nt fragment within the ORF1ab region (YP_009724389.1, positions 11,459-11,858) corresponding to the non-structural protein 6 (Nsp6) (https://www.ncbi.nlm.nih.gov/gene/43740578). Nsp6 is a highly conserved membrane protein with six transmembrane domains, essential for viral replication (36, 37). The alignment with 16,000 consensus sequences of SARS-CoV-2 achieved over 98% identity (https://figshare.com/account/articles/25922725), making the 400 nt fragment an ideal target for ribozyme design. Three ribozymes targeting this region were selected and their specificity were evaluated by aligning target sequences with the genomes of related coronaviruses (SARS-CoV, MERS-CoV, and other common human coronaviruses). The results revealed significant nucleotide sequence differences between SARS-CoV-2 and the other coronaviruses, indicating the specificity of selected ribozymes.

6. The authors indicated that this method is cost-effective and user-friendly. Therefore, the detection cost and preprocessing requirements of this method need to be discussed. 

We appreciate the reviewer's observation. However, the results presented in the manuscript do not aim to evaluate whether the method is cost-effective and user-friendly. These considerations were due to the costs of establishing the methodology and creating prototypes being higher than the cost of manufacturing a commercial kit, mainly in the proof-of-concept and prototyping phase. We estimate that a molecular reaction using HCR-FRET costs about $8-10 under prototype experimental conditions. However, in a manufactured kit, costs would be significantly lower. Therefore, in this manuscript, we emphasize the importance of developing cost-effective and user-friendly methods for long-term global emerging infectious diseases.

We believe that the method described here has potential for the mentioned characteristics based on certain points below. Using a DIY photo-fluorometer prototype allowed us to explore the development of a simple and cost-effective point-of-care detection method, based on digital image analysis. This assertion is supported by comparing the cost of prototyping the device, which can be used without extensive specialized training, with the cost of a spectrofluorometer, or plate-reading spectrofluorometer, and with the specialized knowledge required to operate it. The components and manufacturing cost of the prototype are in the range of a few tens of dollars, compared to the tens of thousands of dollars for a spectrofluorometer, not including the specialized training for the personnel who operate it.

In this context, we revised the manuscript to avoid drawing conclusions about the cost-effectiveness and user-friendliness of the methodology. As previously mentioned in our response to the reviewer's first observation, we have removed the following text from the abstract section (lines 28-32):

These include the need for refrigeration, specialized equipment, and professional expertise. Therefore, there is a need for a diagnostic method that is rapid, cost-effective, user-friendly, and does not require specialized equipment compared to other techniques. Such a method would be extremely useful in resource-limited settings, enhancing readiness for future epidemics and pandemics.

Additionally, in conclusion section we replaced the following text of the manuscript (lines 743-751):

The possibilities of future pandemics, like COVID-19, highlight the critical need for simple and accessibly diagnostic tests to ensure rapid and widespread detection. Investing in user-friendly, cost-effective diagnostic methods is essential for global preparedness and timely response to emerging infectious diseases. Even after the World Health Organization (WHO) declared the end of the COVID-19 public health emergency, the emergence of the EG.5 variant of SARS-CoV-2 has led to new outbreaks, underscoring the importance of sustained vigilance (49). While RT-qPCR remains the gold standard for COVID-19 and other viral diagnoses, our proposed methodology offers a promising alternative that addresses resource limitations, refrigeration challenges, and the need for specialized expertise and equipment. 

by

Considering future pandemics such as COVID-19, it is a necessary to develop simple, cost-effective, user-friendly, accessible, and quickly adaptable diagnostic methods for new viruses and emerging pathogens. In this context, the development of diagnostic methods based on HCR-FRET can result in high-throughput initial screening. In the case of COVID-19, even after the World Health Organization (WHO) declared the end of the COVID-19 public health emergency, the emergence of the EG.5 variant of SARS-CoV-2 led to new outbreaks, highlighting the importance of developing new technologies (56). It is important to emphasize that the proposed methodology is not intended to replace highly specific and sensitive methods based on enzymatic amplification thermocycles, such as PCR and RT-qPCR. Rather, it can serve as a foundation for alternative viral diagnostic methods that address resource limitations and the need for specialized expertise and equipment.

and the following text of the manuscript (lines 752-754):

In this study, a novel molecular method was conceived and developed for detecting SARS-CoV-2 RNA viral based on trans-acting ribozymes and FRET-based hybridization of fluorescent DNA hairpins.

by

In this study, we developed a novel molecular method for detecting viral genome fragments. As a proof of concept, we used SARS-CoV-2 RNA viral fragments. The method is based on trans-acting ribozymes and FRET-based hybridization of fluorescent DNA hairpins.

Other minor corrections and modifications in the manuscript:

1. Line 365-366: 

For the HCR-FRET assay three different final concentrations of the 18 nt initiator DNA molecule …

by

For the HCR-FRET assay, three different concentrations of the 18 nt initiator DNA molecule ….

2. Legenda Figure 3: Line 404-405: 

The average fluorescence spectra emission (n = three independen

---

## [Decision Letter · Decision Letter 1]

27 Aug 2024

A novel viral RNA detection method based on the combined use of trans-acting ribozymes and HCR-FRET analyses

PONE-D-24-22185R1

Dear Dr. Marques Coelho,

We’re pleased to inform you that your manuscript has been judged scientifically suitable for publication and will be formally accepted for publication once it meets all outstanding technical requirements.

Kind regards,

Ruijie Deng

Academic Editor

PLOS ONE

Additional Editor Comments (optional):

Reviewers' comments:

Reviewer's Responses to Questions

**Comments to the Author**

1. If the authors have adequately addressed your comments raised in a previous round of review and you feel that this manuscript is now acceptable for publication, you may indicate that here to bypass the “Comments to the Author” section, enter your conflict of interest statement in the “Confidential to Editor” section, and submit your "Accept" recommendation.

Reviewer #1: All comments have been addressed

Reviewer #2: All comments have been addressed

2. Is the manuscript technically sound, and do the data support the conclusions?

Reviewer #1: Yes

Reviewer #2: Yes

3. Has the statistical analysis been performed appropriately and rigorously? 

Reviewer #1: Yes

Reviewer #2: Yes

4. Have the authors made all data underlying the findings in their manuscript fully available?

Reviewer #1: Yes

Reviewer #2: Yes

5. Is the manuscript presented in an intelligible fashion and written in standard English?

Reviewer #1: Yes

Reviewer #2: Yes

6. Review Comments to the Author

Reviewer #1: (No Response)

Reviewer #2: (No Response)

7. PLOS authors have the option to publish the peer review history of their article (what does this mean?). If published, this will include your full peer review and any attached files.

Reviewer #1: No

Reviewer #2: No

---

## [Editor Report · Acceptance letter]

16 Sep 2024

PONE-D-24-22185R1 

PLOS ONE

Dear Dr. Marques Coelho, 

I'm pleased to inform you that your manuscript has been deemed suitable for publication in PLOS ONE. Congratulations! Your manuscript is now being handed over to our production team.

Kind regards, 

on behalf of

Dr. Ruijie Deng 

Academic Editor

PLOS ONE